# Structural basis for recruitment of peptidoglycan endopeptidase MepS by lipoprotein NlpI

Shen Wang [1], Chun-Hsiang Huang [2], Te-Sheng Lin[1], Yi-Qi Yeh [3], Yun-Sheng Fan[1], Si-Wei Wang[1], Hsi-Ching Tseng[4], Shing-Jong Huang[4], Yu-Yang Chang[1], U-Ser Jeng[3], Chung-I Chang [5] & Shiou-Ru Tzeng [1] ✉

Peptidoglycan (PG) sacculi surround the cytoplasmic membrane, maintaining cell integrity by withstanding internal turgor pressure. During cell growth, PG endopeptidases cleave the crosslinks of the fully closed sacculi, allowing for the incorporation of new glycan strands and expansion of the peptidoglycan mesh. Outer-membrane-anchored NlpI associates with hydrolases and synthases near PG synthesis complexes, facilitating spatially close PG hydrolysis. Here, we present the structure of adaptor NlpI in complex with the endopeptidase MepS, revealing atomic details of how NlpI recruits multiple MepS molecules and subsequently influences PG expansion. NlpI binding elicits a disorder-to-order transition in the intrinsically disordered N-terminal of MepS, concomitantly promoting the dimerization of monomeric MepS. This results in the alignment of two asymmetric MepS dimers respectively located on the two opposite sides of the dimerization interface of NlpI, thus enhancing MepS activity in PG hydrolysis. Notably, the protein level of MepS is primarily modulated by the tail-specific protease Prc, which is known to interact with NlpI. The structure of the Prc-NlpI-MepS complex demonstrates that NlpI brings together MepS and Prc, leading to the efficient MepS degradation by Prc. Collectively, our results provide structural insights into the NlpI-enabled avidity effect of cellular endopeptidases and NlpI-directed MepS degradation by Prc.

Peptidoglycan (murein) is a crucial structure that wraps around the cell membrane, protecting it from osmotic pressure[1–5]. The general structure of peptidoglycan (PG) sacculus is composed of multiple linear glycan strands, which are made of alternating β1,4-linked *N*-acetylglucosamine (NAG) and *N*-acetylmuramic acid (NAM) covalently linked to the first residue of the short peptide chain comprising 2–5 amino acids[2–9]. The most common interconnection bridging neighboring peptide chains is the 4−3 cross-link between D-Ala and *meso*-diaminopimelic acid (DAP), leading to the formation of a net-like structure to protect the cells from osmotic rupture. Compared to the 4−3 cross-link, the 3−3 cross-link is infrequently observed[10–13]. During cell growth, fully closed sacculi need to be cleaved to incorporate new murein strands for expansion of the peptidoglycan mesh[2,14–16]. Several PG endopeptidases, which include MepS (*spr*), MepM (*yebA*), MepH (*ydhO*), MepK, PBP4 (*dacB*), PBP7 (*pbpG*), MepA and AmpH[14–25], have been identified to be involved in the growth and survival of

[1]Institute of Biochemistry and Molecular Biology, College of Medicine, National Taiwan University, Taipei, Taiwan. [2]Protein Diffraction Group, Experimental Facility Division, National Synchrotron Radiation Research Center, Hsinchu, Taiwan. [3]Soft Matter Science Group, Scientific Research Division, National Synchrotron Radiation Research Center, Hsinchu, Taiwan. [4]Instrumentation Center, National Taiwan University, Taipei, Taiwan. [5]Institute of Biological Chemistry, Academia Sinica, Taipei, Taiwan. ✉e-mail: srtzeng@ntu.edu.tw

bacteria[2,14,16,26–32]. In *Escherichia coli*, three out of the eight endopeptidases – MepS, MepM, and MepH - are redundantly essential for cell wall expansion, while the mutant lacking all three murein endopeptidases leads to the formation of abnormally deformed cells, which ultimately undergo cell lysis[14]. Recent studies have revealed that MepS, MepM, and MepH hydrolyze the 4−3 cross-links for PG enlargement[14], and MepM, MepK, and MepA are the reported endopeptidases with the ability to cleave the 3−3 cross-links, resulting in enhancing the PG strand incorporation sufficiently[13,17,25].

NlpI, an outer membrane-anchored lipoprotein, is ubiquitously distributed throughout the entire cell envelope and is primarily found in Gram-negative bacteria[33]. By adopting the structure of tetratricopeptide repeats (TPRs)[34] that mediate protein-protein interactions, NlpI is known to play multiple roles, including cell division, cell wall metabolism, virulence, and association with host cells[35–38]. As an adaptor protein, NlpI has been found to interact with different classes of hydrolases and further associate with PG synthetic machinery[39,40]. Changes in NlpI levels alter the quantity and stability of major envelope biogenesis components[35,39]. NlpI respectively interacts with three endopeptidases MepS, MepM, and PBP4 with similar apparent $K_D$ (100–200 nM); however, PBP7 has a more complicated binding mode with an EC50 value of 422 nM[39]. The adaptor NlpI helps localize NlpI-interacting endopeptidases, including MepS, MepH, MepM, MepK, PBP4, and PBP7. In vitro experiments showed that NlpI enhances the activity of MepS against muropeptides, while the activity of MepM is significantly reduced in the presence of NlpI. Notably, NlpI can cause different endopeptidases to form various trimeric complexes, such as MepS-NlpI-PBP4 and MepS-NlpI-PBP7, as these two endopeptidases cannot interact directly. The interactions of NlpI with certain hydrolases contribute to the localization and modulation of PG hydrolysis and elongation[41]. This suggests that NlpI plays a role in bringing endopeptidases close to the complexes involved in peptidoglycan synthesis, thereby spatially connecting PG hydrolysis to PG expansion[39,40]. Reconstitution experiments demonstrated that NlpI promotes the organization of PG multienzyme complexes[39], which include LpoA (a synthesis regulator), PBP1A (a bifunctional synthase), NlpI and endopeptidases, suggesting that the integration of hydrolases and synthases during PG expansion is conceivably facilitated by NlpI[39,40].

Among these endopeptidases, MepS is found to be quite abundant during the log phase of cell growth but declines abruptly during the stationary phase[42]. The protein level of MepS is known to be regulated by the periplasmic PDZ-protease Prc (or tsp, tail-specific protease) in complex with the adaptor NlpI[42]. Previous reports indicate that Prc cannot efficiently degrade MepS in the absence of NlpI[42,43], indicating the vital role of NlpI in MepS recruitment. Indeed, the degradation of MepS is markedly dependent on the adaptor NlpI, which recognizes MepS and presents it to the tail-specific protease Prc. The mutants lacking Prc or NlpI lose the ability to regulate the level of MepS, resulting in a significant increase in the MepS protein throughout the distinct phases of bacterial growth. This leads to the development of long filaments and growth defects in a low-osmolarity environment[33,44].

The structure of Prc-NlpI complex [Protein Data Bank (PDB) ID 5WQL] was solved at 2.3 Å by X-ray crystallography[43], revealing that a symmetric NlpI homodimer is attached to two bowl-shaped Prc proteins located on each side of dimeric NlpI. The adaptor protein NlpI contains four TPRs and the TPR2 of NlpI mainly interacts with helices h1 and h14 of Prc, namely the NlpI-interaction domain, through an extensive electrostatic network. The unliganded PDZ domain of Prc adopts a misaligned active site in a default resting conformation, while the ligand-bound PDZ domain results in rearrangement of the active site[45], thus activating the proteolytic activity of Prc. Based on the functional features of the tail-specific protease Prc, the MepS molecules are assumedly placed with a particular orientation in the interspace between Prc and NlpI[43].

Our current understanding of the underlying molecular basis is still incomplete due to the lack of structural information on NlpI-endopeptidases and Prc-NlpI-endopeptidase complexes. The structures of the endopeptidases MepS and PBP4 were respectively solved by nuclear magnetic resonance (NMR) spectroscopy and X-ray crystallography (PDB IDs 2K1G and 2EX2)[46,47]. Here, we report the crystal structures of NlpI-MepS and Prc-NlpI-MepS complexes determined at 2.8 and 3.5 Å resolution, respectively. The structure of NlpI in complex with MepS provides molecular insight into NlpI-interacting endopeptidase colocalization, while the Prc-NlpI-MepS complex elucidates the molecular mechanism of NlpI-mediated MepS degradation by the tail-specific protease Prc. Combining structural and mutational analyses with biophysical and biochemical assays, our results reveal previously unidentified insights into NlpI-induced endopeptidase colocalization and NlpI-mediated MepS degradation by the PDZ-protease Prc.

## Results

### The intrinsically disordered N-terminal of the endopeptidase MepS is involved in the interaction with adaptor NlpI

The molecular details of the role of MepS in the formation of the heterocomplex with the adaptor NlpI are still unclear. Based on a docking model with mutational analysis, a model is proposed on the notion that only the structured domain plays a primary role in complex formation[42,43]. The N-terminal segment of MepS was not included in previously published NMR and X-ray structure studies owing to the significant disorder noted during NMR spectral screening[46,48]. Whether the disordered region of MepS takes part in MepS-NlpI heterocomplex formation has never been examined. We therefore chose to perform NMR titration experiments using the soluble mature form of MepS (mMepS) and the truncated mutant dN36-MepS, which is devoid of residues 2 to 36 (as illustrated in Fig. 1A). Both proteins were expressed and purified without the lipoprotein signal peptides[34,43,46]. The NMR spectra of dN36-MepS (residues 37–162) and mMepS (residues 1–162) exhibited a high degree of superimposition (Fig. 1B), except for residues 37–45, 69, 81, 105, and 160 (Supplementary Fig. 1A). Approximately 90% of mMepS backbone resonances were assigned through multidimensional heteronuclear NMR experiments (Supplementary Fig. 1B). Using NMR resonances of the backbone atoms, the δ2D algorithm[49] was employed to determine the secondary structure populations of mMepS. The analysis revealed diminished values in the secondary structure propensities for residues 6–38, with ~3% α-helix, ~5% β-strand, and ~25% PPII (Supplementary Fig. 1C), confirming the lack of structural ordering in the N-terminal region of mMepS.

Notably, the NMR spectra of $^{15}$N-labeled dN36-MepS in the absence and presence of unlabeled NlpI, which lacked the lipoprotein signal peptide, were very similar (Fig. 1C), while the peak signals of mMepS almost disappeared or broadened beyond detection upon addition of NlpI (Fig. 1D). The reduction in peak intensity was quantified through two-dimensional $^{15}$N-$^1$H NMR spectroscopy, which revealed signal intensity ratios of 77% ± 6% for dN36-MepS and less than 10% for mMepS (Supplementary Figs. 2 and 3). This suggests that the disordered N-terminal region of mMepS greatly contributes to its binding to NlpI. Afterwards, we tested the importance of the disordered N-terminal region by preparing the truncated mutants MepS-N39 (residues 1–39) and MepS-N53 (residues 1–53) as illustrated in Fig. 1A. The spectra of MepS-N39 and MepS-N53 exhibited highly disordered peptide segments (Fig. 1E), as indicated by rather intense peaks with poor dispersion at approximately 7.5–8.5 ppm of the proton dimension. Titration of unlabeled NlpI into $^{15}$N labeled MepS-N39 and MepS-N53, respectively, caused substantial peak broadening (Fig. 1F). The findings confirmed that even in the absence of the core structure, the disordered N-terminal of MepS maintains its binding capacity for the adaptor protein NlpI, thus substantiating its role in the formation of the NlpI:MepS heterocomplex.

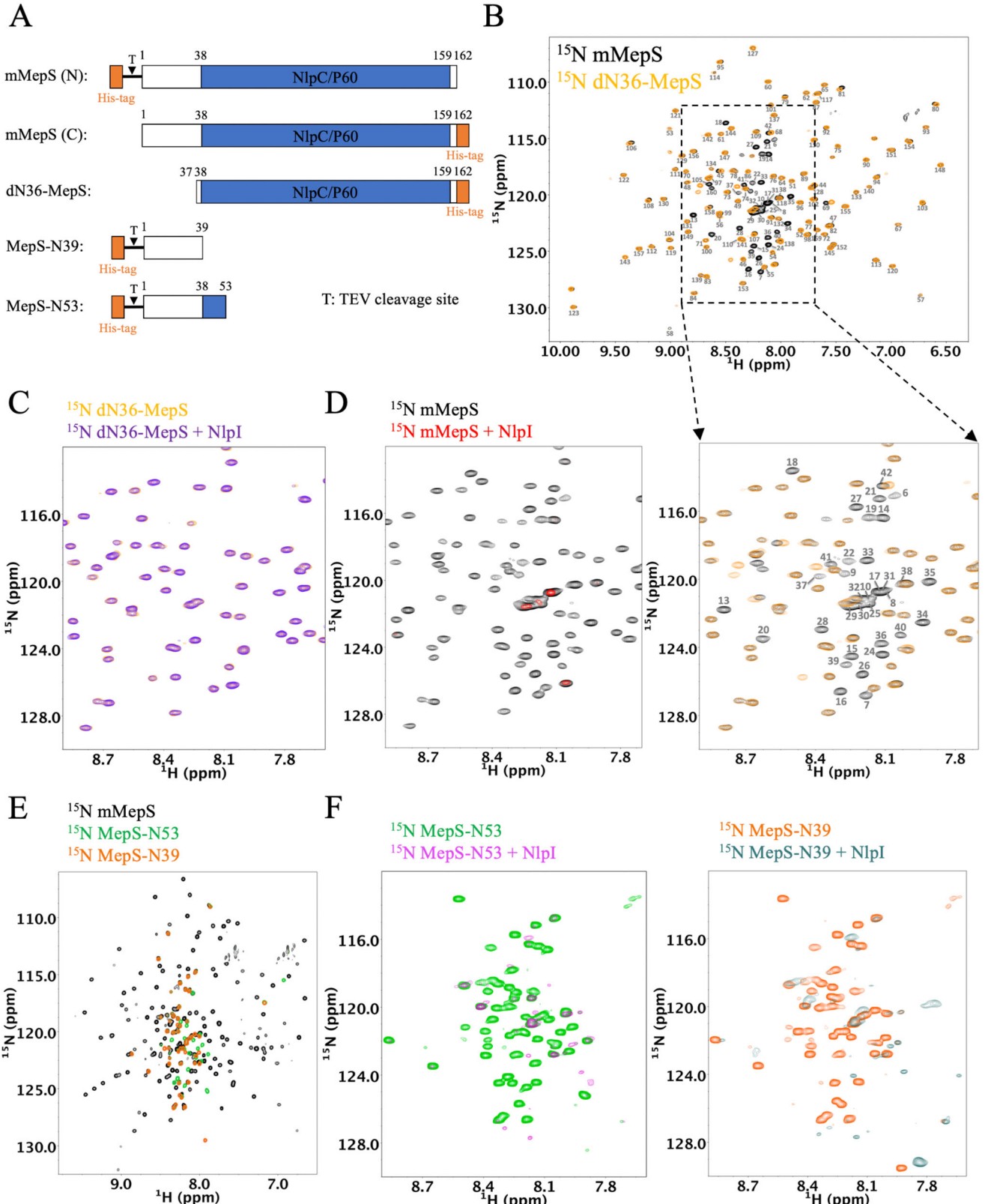

Previous research has shown that dN36-MepS remains stable as a monomeric form even at high concentrations (~1 mM)[46]. We performed ¹H-detected 1D ¹⁵N relaxation experiments to measure the average ¹⁵N $T_1$ and $T_2$ relaxation times for mMepS and dN36-MepS[46,50], respectively. To minimize contributions from the disordered regions of mMepS, ¹⁵N $T_1$ and $T_2$ values of mMepS and dN36-MepS were respectively extracted by the decay of the integrated ¹Hᴺ intensity

between 9.0 and 9.6 ppm (Supplementary Fig. 4A, B). The $\tau_c$ values for mMepS and dN36-MepS were determined to be 8.7 and 8.5 ns, respectively, suggesting that both proteins exist in a monomeric form. Furthermore, we applied heteronuclear NOE experiments to investigate the local backbone flexibility of mMepS at high concentrations, and the values of heteronuclear NOE were decreased for the peaks from the N-terminal of MepS (Supplementary Fig. 5), indicating that

**Fig. 1 | NMR characterization of interactions of NlpI with various mMepS protein constructs. A** Schematic representation of MepS constructs utilized for investigating the role of the N-terminal region. The mature full-length construct (mMepS) consists of the N-terminal disorder region (1–37, shown in white) and the C-terminal NlpC/P60 domain (shown in blue). The dN36-MepS mutant lacks the 36 amino acids at the N-terminal while MepS-N39 and MepS-N53 contain only the N-terminal 39 and 53 residues, respectively. T denotes the TEV cleavage site. The lipidation site of MepS at residue C1 in its mature form is replaced with Methionine in all MepS constructs. **B** Comparison of the $^1$H-$^{15}$N TROSY-HSQC spectra of mMepS (in black) and dN36-MepS (in yellow) obtained at 25 °C. The cross-peaks are annotated with the residue numbers of mMepS. The overlay of the expanded region of the 2D spectrum highlights the natively disordered N-terminal of mMepS. **C** $^1$H-$^{15}$N

TROSY-HSQC spectra of dN36-MepS (50 μM) were acquired in both the absence (yellow) and presence (purple) of the unlabeled NlpI dimer (25 μM). Analysis using two-dimensional $^1$H-$^{15}$N NMR spectroscopy reveals a reduction in peak intensity. **D** $^1$H-$^{15}$N TROSY-HSQC spectra of mMepS (50 μM) were acquired in both the absence (black) and presence (red) of unlabeled NlpI dimer (25 μM). **E** Overlaying the $^1$H-$^{15}$N HSQC spectra of mMepS (in black), MepS-N53 (in green), and MepS-N39 (in orange) acquired at 25 °C. **F** The spectral data display the cross-peaks for both MepS-N53 and MepS-N39 in the presence of NlpI, represented by magenta and cadet blue colors. The MepS-N39 and MepS-N53 truncated mutant samples were prepared without TEV protease treatment, and the less affected peaks are mainly from the His-tag and TEV sites.

the flexibility on ps-ns timescale of this region is prominently increased. To further compare the interactions of NlpI with mMepS and dN36-MepS, we performed the size-exclusion chromatography (SEC) experiments to examine the samples containing three proteins (NlpI, mMepS, and dN36-MepS), revealing that the coelution of NlpI with mMepS but not with dN36-MepS (Fig. 2A). Sodium dodecyl sulfate–polyacrylamide gel electrophoresis (SDS–PAGE) of the SEC peak fractions confirmed the presence of NlpI:mMepS complex, as dN36 eluted at the second peak at 17 ml (Fig. 2A). The formation of a stable complex between NlpI and mMepS was observed, with both coeluting at a peak volume of approximately 13 ml, while NlpI, mMepS, and dN36-MepS separately eluted at peak volumes of approximately 14, 16, and 17 ml (Fig. 2B).

**The structure of NlpI:mMepS complex**

To characterize the basis of how the lipoprotein NlpI mediated the endopeptidase colocalization near the outer membrane, we determined a 2.8 Å resolution structure of NlpI in complex with mMepS by X-ray crystallography. The structure revealed a heterohexameric complex with a homodimer of NlpI bound to four mMepS molecules (Fig. 3A and Supplementary Table 1). We also performed size exclusion chromatography coupled with small-angle X-ray scattering (SEC-SAXS) experiments to examine the stoichiometry of the complex structure of NlpI-mMepS. The analysis revealed that the SAXS data acquired for the NlpI-mMepS complex resulted in an Rg of 35.1 Å. Moreover, the SAXS profile of NlpI-mMepS is consistent with the crystallographic results, with $\chi^2 = 1.73$, confirming that the hexameric structure does exist in solution (Fig. 3B). In the crystal structure, NlpI exhibited the expected four tetratricopeptide repeats (TPRs) in mMepS-bound states (Supplementary Fig. 6), in agreement with the published apo structure (with a mean pairwise rmsd of 0.356 Å over 386 Cα atoms). Surprisingly, the intrinsically disordered N-terminal of monomeric mMepS becomes folded and dimerized upon NlpI binding (Fig. 3C and Supplementary Fig. 7), thereby guiding four mMepS to form two asymmetric dimers bound to a homodimeric NlpI (namely NlpI and NlpI′). One protomer (mMepS-1) of NlpI-bound mMepS has the extended α1 (residues Q28 to W51), while the 12-residue segment comprising Q28 to D39 lacks a defined secondary structure in its apo state (Fig. 3D and Supplementary Fig. 7). This segment simultaneously engages in extensive interactions with the h1 of NlpI and the h3′ of NlpI′ located at the dimerization interface of NlpI, thus contributing to the extension of α1 by approximate 3.3 turns and 18 Å (Fig. 3D and Supplementary Fig. 8). Furthermore, the N-terminal region of the other protomer (mMepS-2) also synchronously makes extensive contacts with the h1-h2 of NlpI and the extended α1 of mMepS-1, thereby facilitating the formation of a new α0 comprising residues 23–35 (approximate 3.6 turns).

By contrast, the core structure of asymmetric dimeric mMepS in the NlpI-bound state is highly similar to that of monomeric apo mMepS, which contains a conventional papain-like α + β fold (Fig. 3D). The core structure of mMepS-1 mainly interacts with that of mMepS-2 (Supplementary Fig. 9), while the extended α1 of mMepS-1 engages the

interface of NlpI dimer. Notably, mMepS-2 of the complex structure simultaneously engages mMepS-1 and one protomer NlpI but not the other NlpI′ (Supplementary Fig. 10). In brief, mMepS and NlpI form a hetero-hexameric assembly with structured and disordered mMepS regions (Fig. 3A). We conclude that the colocalization of four mMepS molecules can be mediated by a NlpI homodimer, suggesting that multiple mMepS may work cooperatively and thus enhance the avidity of mMepS-PG interaction.

To validate the above structural observations, we measured the affinity of NlpI for the mMepS mutants by isothermal titration calorimetry (ITC) experiments. In the case of NlpI binding to mMepS (Fig. 4A and Supplementary Table 2), we obtained the enthalpy-driven interaction with a $K_D$ of $0.24 \pm 0.02$ μM, which was consistent with previously published results[43]. Compared with mMepS, the NlpI titration into the truncated dN36-MepS mutants did not cause an enthalpically favorable binding reaction, suggesting that the initial 36 N-terminal residues significantly contributed to NlpI binding (Fig. 4B and Supplementary Table 2). By contrast, we investigated the affinity of the truncated mutants MepS-N53 and MepS-N39 for NlpI; the results revealed that both interactions were enthalpically favorable, with the $K_D$ values of $1.45 \pm 0.22$ and $2.68 \pm 0.15$ μM, respectively, but with the 6- to 11-fold differences in binding affinity (Fig. 4C, D and Supplementary Table 2). We also chose to mutate critical NlpI–mMepS interacting residues inferred from our X-ray ternary structure. For example, the hydrophobic side-chain of L24 of mMepS-1 has multiple contacts with the side-chains of R78′, A79′, R82′, Q108′ and A109′, and the aromatic ring of Y105′ located at the h3′-h4′ of NlpI′; mMepS-L24R binds to NlpI dimer with an approximate 28-fold increase in the $K_D$ value (Fig. 4E and Supplementary Table 2). The side-chain of mMepS-1 Q28 has contacts with the side-chains of residues R82′, N83′, and S86′ located at the h3′ of NlpI′; mMepS-Q28A resulted in the decreased binding affinity by a factor of 7.7 (Supplementary Fig. 11A and Table 2). The aromatic ring of mMepS-1 F31 has hydrophobic contacts with NlpI h1 (involved in the side-chains of L38, V42 and A45) and NlpI′ h3′ (involved in the side-chains of A79′ and N83′) located at the dimerization interface of NlpI homodimer. Interestingly, the NlpI titration to mutation F31A had no sufficient heat release or absorption to allow the $K_D$ determination (Fig. 4F and Supplementary Table 2), indicating that the bulky hydrophobic side-chain of F31 plays a dominant role in the NlpI–mMepS interaction. The side-chain of mMepS-1 D39 has polar contacts with residue R46 of NlpI h1; the $K_D$ of mMepS-D39A variant for NlpI was almost unchanged ($0.39 \pm 0.11$ μM), and there seems to be enthalpy-entropy compensation that leads to very little or no effect on the free energy changes (Supplementary Fig. 11B and Supplementary Table 2). In summary, we conducted ITC experiments to examine the binding of NlpI with MepS mutants, including dN36-MepS, N53, N39, MepS-L24R, MepS-Q28A, MepS-F31A and MepS-D39A. The enthalpic contribution is notably influenced by the N-terminal of MepS, where the residues (Q28 and D39) involved in the hydrophilic interactions result in little or no difference in the binding affinity, while the hydrophobic residues (L24 and F31) are the main determinants of the association with NlpI.

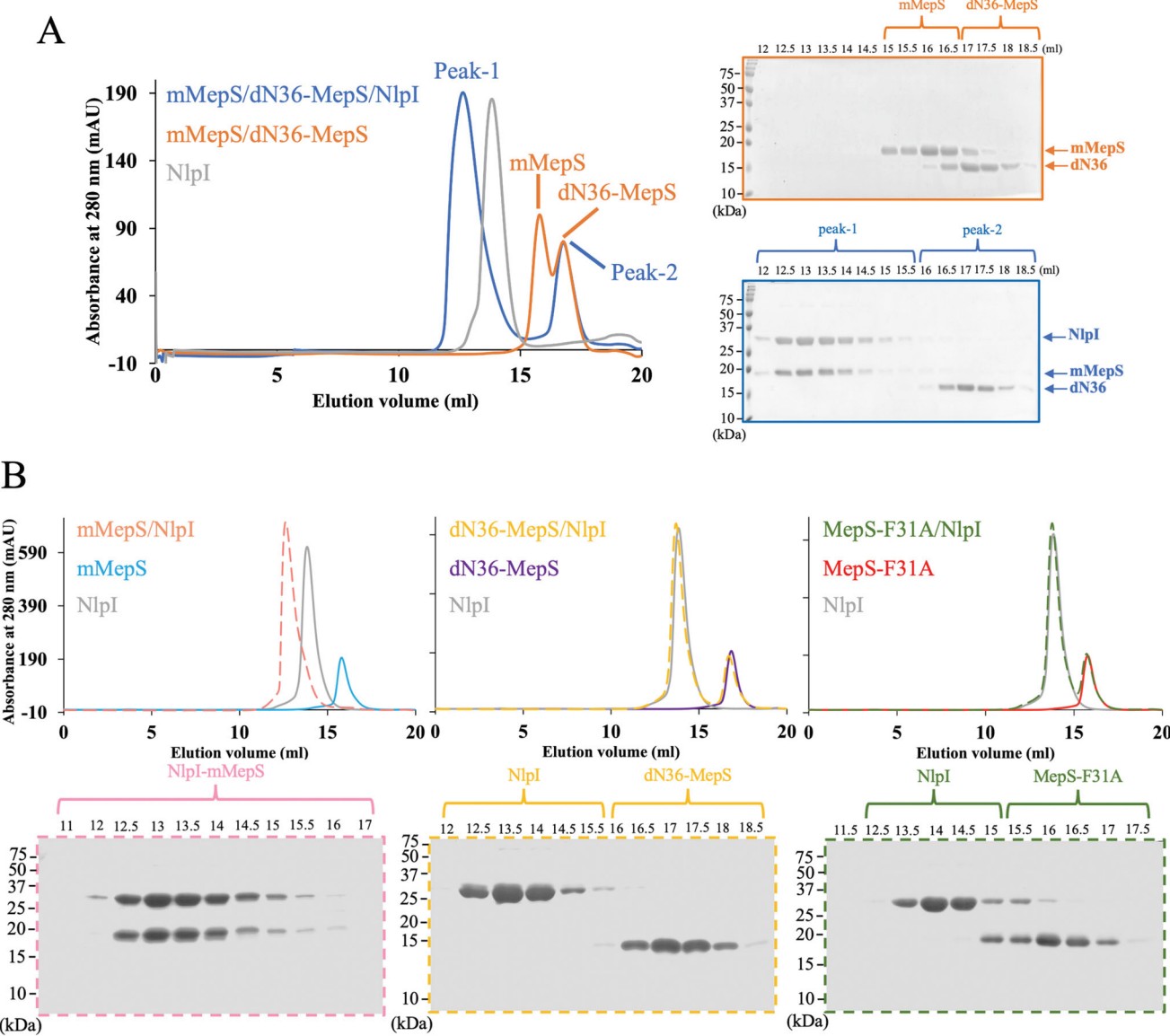

**Fig. 2 | Exploring the involvement of the disordered N-terminal region of mMepS in the formation of the NlpI-mMepS complex via size exclusion chromatography. A** The gel filtration profile of a sample containing equimolar concentrations of mMepS, dN36-MepS, and NlpI is depicted in blue. Subsequent analysis using Coomassie-blue stained SDS-PAGE gel on the peak fractions. The elution profiles of NlpI dimer alone and the sample containing two proteins (mMepS and dN36-MepS) are represented in grey and orange, respectively. **B** Purified NlpI was mixed with mMepS (pink), dN36-MepS (yellow), or MepS-F31A (green) and then analyzed using gel filtration chromatography to assess the

formation of complexes. Dashed lines were employed to delineate distinct protein mixtures, enhancing the clarity and visibility of each line. All experiments were conducted using the Superdex200 10/300 GL column (GE Healthcare), and elution peaks were monitored through UV absorbance at 280 nm. The resulting SEC elution profiles were further confirmed by Coomassie-stained SDS-PAGE gels, with each gel corresponding to specific peak fractions and distinguished by different colors. The data represent a single experiment out of three independent ($n = 3$) measurements. Source data are provided as a Source Data file.

Having identified the mutants F31A and dN36 involved in the loss of the exothermic reactions using ITC experiments, we performed SEC experiments to investigate the interactions of NlpI with mMepS mutants. The SEC chromatograms of both F31A and dN36 mutants confirmed that disrupting the interaction between the N-terminal of MepS and NlpI compromised the NlpI-mMepS assembly as the F31A and dN36 mutants were both eluted as separate peaks in the presence of NlpI (Fig. 2B). On the contrary, the NlpI:mMepS complex was stably formed as demonstrated by the shifted protein peak with a larger apparent size (Fig. 2B).

We also utilized NMR spectroscopy to investigate the interactions between NlpI and mMepS variants. Of particular interest was the unexpected impact of the F31A mutation on NlpI binding, as revealed

by calorimetric analysis. Even though we observed changes in the NMR signals of the $^{15}$N-labeled F31A variant upon exposure to increasing concentrations of the unlabeled NlpI dimer (Fig. 4G), the effect was significantly less significant compared to the impact seen in the mMepS-NlpI interaction (Supplementary Fig. 3). This is particularly evident in the N-terminal region, where intensity ratios ($I/I_O$) were higher than the average plus one standard deviation, attributable to the F31A point mutation. The result highlighted the crucial role of the hydrophobic aromatic ring of F31 in the interaction with NlpI. With the titration of increasing NlpI dimer, the NMR peaks of the L24R variant were significantly broadened or disappeared, except for a few residues near L24R (Supplementary Fig. 12). These findings collectively suggest that the hydrophobicity of crucial N-terminal residues in mMepS

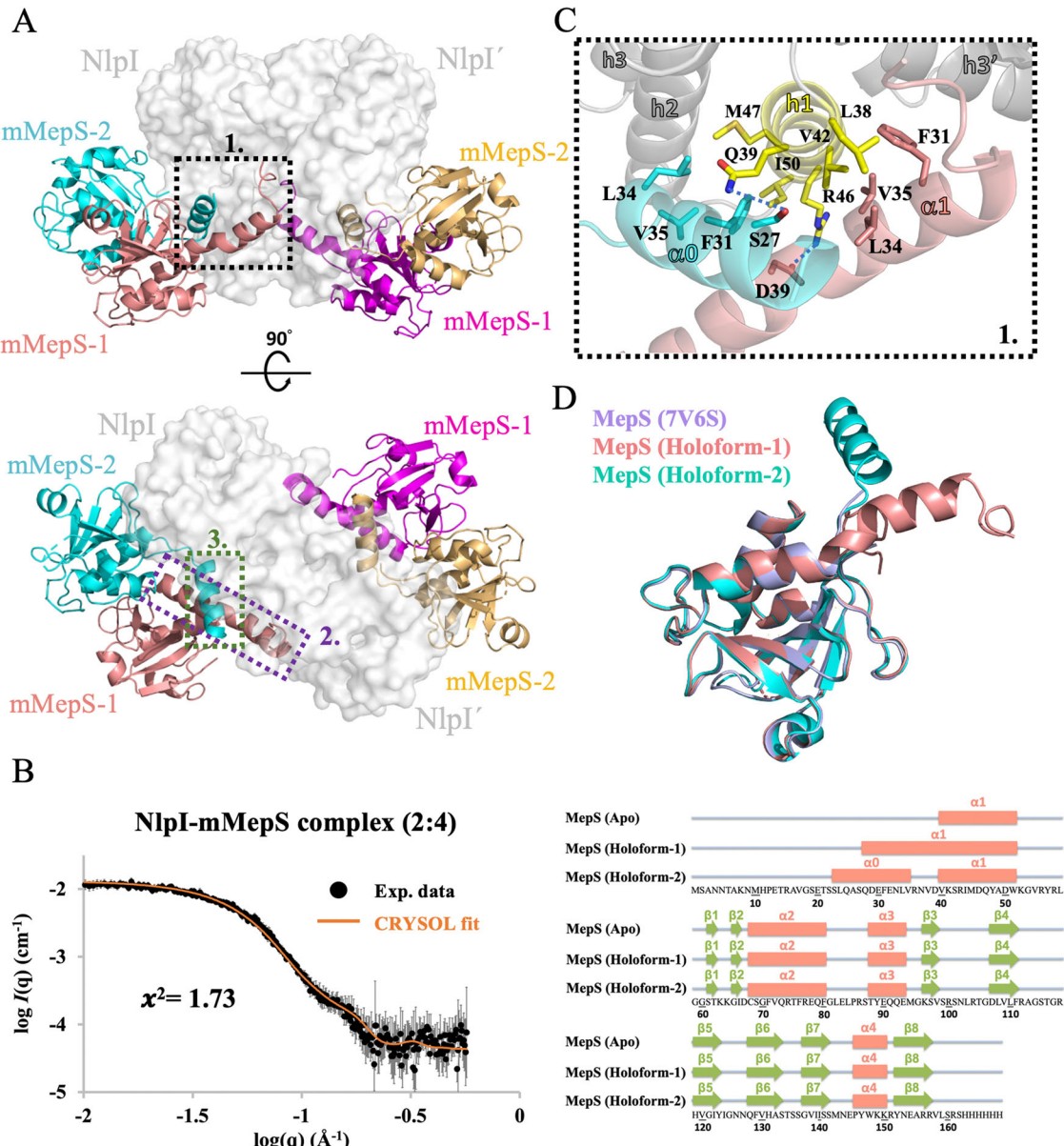

**Fig. 3 | Structural insights into the NlpI·mMepS complex by crystal structure and SAXS. A** Two orthogonal views of the NlpI homodimer complexed with mMepS are depicted with the indicated location of the interaction sites, highlighting the binding interfaces between the proteins. The detailed interactions highlighted in dashed boxes 1 to 3 are further described in Fig. 3C and Supplementary Fig. 8. The surface of the NlpI dimer is rendered in grey, while the bound mMepS proteins are represented in cartoon form using colors of salmon, cyan, magenta, and golden. This color scheme is consistently used throughout the entire document. **B** SAXS analysis was conducted on the NlpI·mMepS complex, and the results were compared with the theoretical SAXS curve calculated from the crystal structure of the complex. The superimposition of the experimental size exclusion chromatography coupled with small-angle X-ray scattering (SEC-SAXS) profile in black and the theoretical SAXS curve in orange showed alignment, with a χ2 value of 1.73, confirming the presence of the hexameric structure in solution. The error

bars represent the root-mean-square deviations of the measured scattering intensity at each scattering vector (**q**) value. Source data are provided as a Source Data file. **C** The interactions between NlpI and mMepS proteins involve specific hydrophobic contacts. The hydrophobic side chains of L24, F31, L34, V35, and V38 in the extended alpha-1 region of mMepS-1 form hydrophobic contacts with residues L38, V42, I43, and A45 of NlpI h1, as well as residues R78'-L80', Y105', and Q108' located at NlpI' h3'-h4'. **D** Secondary structure comparison of mMepS in the context of apo state and NlpI-bound states. The apo form of mMepS features an intrinsically disordered N-terminal preceding the core domain, as characterized by our NMR chemical shift assignments (BMRB ID 51949). The α helices are depicted as salmon rectangles, while the β strands are indicated by green arrows. The secondary structural elements of NlpI-bound mMepS and the apo form of mMepS (PDB ID 7V6S) are shown above the protein sequence, with the α helices depicted as salmon rectangles, and green arrows indicating the β strands.

greatly contributes to NlpI binding, leading to a disorder-to-order transition during the binding event.

In accordance with the NlpI-mMepS complex structure, the N-terminal region of one protomer (mMepS-1) prominently engages with the dimerization interface of NlpI, while that of the other protomer (mMepS-2) only interacts with one of NlpI. This suggests that the dimerization interface of NlpI is crucial for the binding of mMepS-1 but

not mMepS-2 (Supplementary Fig. 8). To explain the stoichiometry of 1 obtained in the ITC experiments, we used a monomeric mutant NlpI-ΔN (Supplementary Fig. 13A) to characterize the interactions by NMR and ITC experiments. The NlpI-ΔN (37–294) mutant, which lacks residues 1–36, is unable to form a dimer and adopts a monomeric conformation[43]. The SEC data for dimeric NlpI (residues 20–294) and monomeric NlpI-ΔN (residues 37–294) are presented in grey and

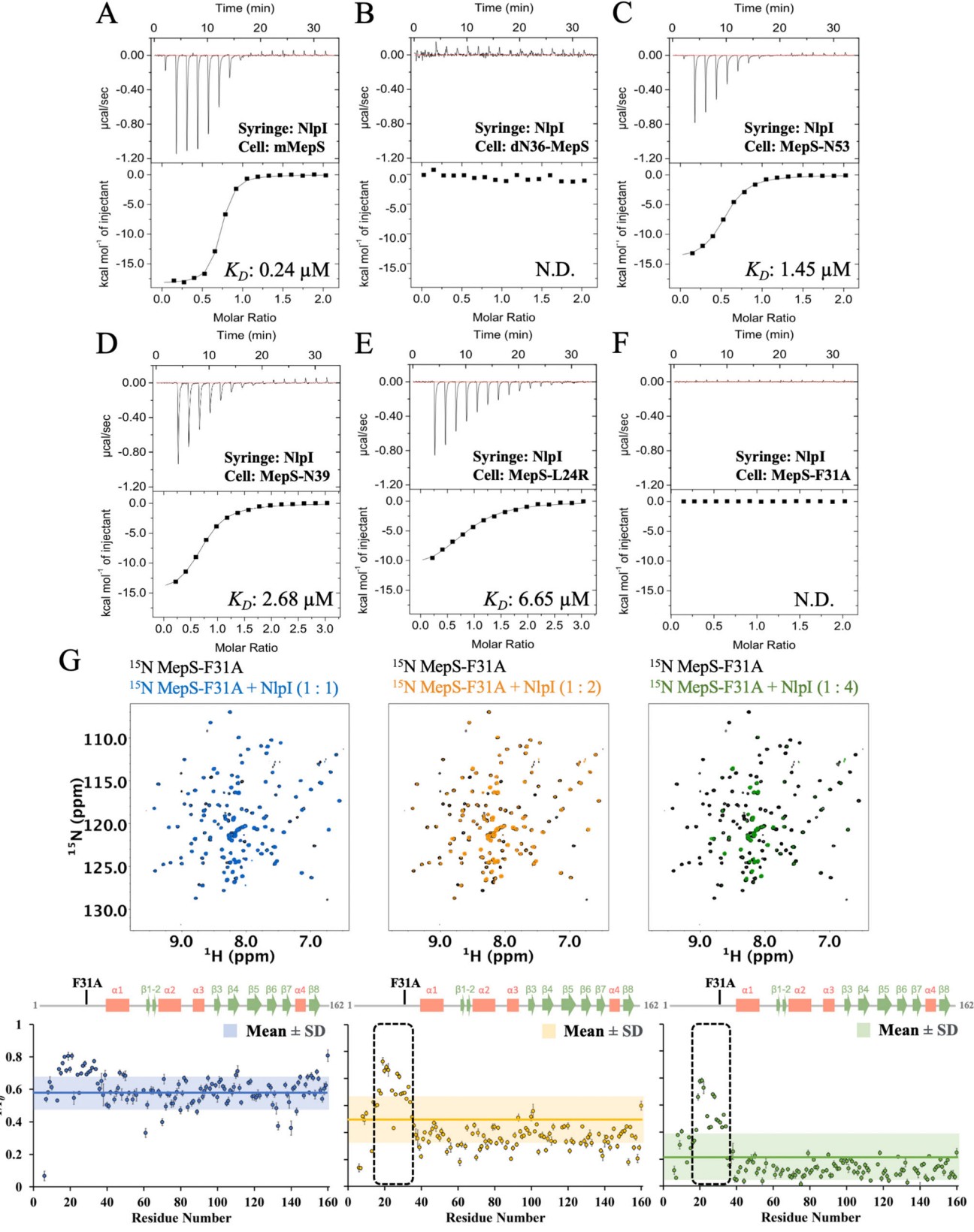

purple, respectively (Supplementary Fig. 13A). Titration of unlabeled NlpI-ΔN into ¹⁵N-labeled mMepS causes line-broadening from residues 26 to 162 (Supplementary Fig. 13B). This result aligns with the crystal structure, as residues 23-25 engage with NlpI dimer. The interaction between mMepS and NlpI-ΔN did not exhibit exothermic or endothermic signals as monitored via ITC (Supplementary Fig. 13C), while

the binding of NlpI-ΔN to mMepS was directly observed by NMR experiments. Therefore, we propose that only mMepS-1 interacts with the NlpI dimer with an enthalpically favorable binding response as mMepS-2 could bind to NlpI without apparent heat release or absorption; therefore, a lower stoichiometry was observed in the ITC binding reaction.

**Fig. 4 | Structure-derived mutants of mMepS at the binding interface perturb interaction with NlpI in vitro. A–F** The ITC fitting curves illustrate the interactions between wild-type mMepS and MepS mutants with NlpI. The presented raw data, post-baseline correction (top), along with the corresponding binding isotherms (bottom), depict the integrated heat peaks plotted against the molar ratio of ligands. The data represent a single experiment out of three independent ($n = 3$) measurements and the results are presented as the mean values ± standard deviations. The $K_D$ values are labeled and detailed in Supplementary Table 2. The designation "N.D." indicates that it is not determined under our experimental conditions. **G** green, respectively). Titrations of MepS-F31A with NlpI are illustrated at 1:1 (blue), 1:2 (orange), and 1:4 (green) ratios. The ratio of NMR signal intensities, denoted as $I/I_O$, refers to the signal intensity of NlpI-bound MepS-F31A ($I$) relative to that of free MepS-F31A ($I_O$). The average intensity ratio for each titration is displayed using its corresponding color code, while the ranges of the mean values ± one standard deviation are depicted with shading. The dashed-line box identifies residues exhibiting intensity ratios above the average plus one standard deviation, attributed to the point mutation F31A. Additionally, the secondary structure of mMepS is indicated above for reference. Data of one representative experiment performed on $n = 2$ biologically independent samples. The lack of intensity ratios in some residues could result from proline residues, unassigned residues, or residues with overlapping signals, potentially affecting measurement accuracy. The error bar reflects the uncertainty derived from the peak's signal-to-noise ratio. Source data are provided as a Source Data file.

## Structural analysis of the Prc$^{SK}$-NlpI-mMepS complex

The proteolytic regulation of mMepS is dependent on the tail-specific protease Prc, and the adaptor NlpI greatly facilitates the degradation process. We used NMR to detect the interaction of $^{15}$N-labeled mMepS with unlabeled Prc$^{SK}$, revealing that the peaks from mMepS were largely unaffected and showed 93% ± 11% peak intensity in the presence of the same concentration of Prc$^{SK}$ (Supplementary Fig. 14). The NMR signal of mMepS remained at 91% ± 11% even after titration with a two-fold excess of unlabeled Prc$^{SK}$, suggesting that no substantial binding occurred between mMepS and Prc$^{SK}$. The result was in agreement with previous reports, indicating that mMepS is inaccessible for Prc$^{SK}$ binding. To understand how NlpI enhances the efficiency of mMepS proteolysis by Prc, we determined the crystal structure of Prc$^{SK}$-mMepS-NlpI at 3.5 Å, revealing that two monomeric Prc$^{SK}$ binds to heterohexameric NlpI-mMepS to form a 2:2:4 heterooctameric complex (Fig. 5A). To exclude the possibility that our crystals contained incomplete or degraded Prc$^{SK}$ proteins during the crystallization process, several crystals of Prc$^{SK}$-NlpI-mMepS were dissolved and examined via SDS-PAGE analysis, demonstrating the Prc$^{SK}$ variant in our crystals was intact (Supplementary Fig. 15). We also examined the stoichiometry of the Prc$^{SK}$-mMepS-NlpI complex through simultaneous SAXS and UV–Vis absorption measurements[51]. The results showed that two monomeric Prc$^{SK}$ molecules bind to heterohexameric NlpI-mMepS, forming a 2:2:4 hetero-octameric complex measured at the eluent peak, with a confidence level exceeding 90% (Fig. 5B and Supplementary Note 1).

The structure of NlpI-mMepS in the Prc$^{SK}$-bound state closely resembles that in the absence of Prc$^{SK}$, with a rmsd of 1.31 Å for the backbone, showing that a dimeric NlpI symmetrically bound to two asymmetric mMepS dimers, as described in Fig. 3 and Supplementary Fig. 7. Although the side-chain orientations can be easily defined in both NlpI and mMepS (Supplementary Fig. 16), the density maps of Prc are quite poor, except for the residues 42–120 (the N-terminal of NHD), 208–244 (the vault region) and 479–520 (the partial platform domain). The structure of Prc in the NlpI-mMepS bound state still forms a bowl-like body (Supplementary Fig. 17), which is superimposed on that of Prc-K477A in the NlpI-bound state. The NlpI-interaction domain comprising helix h1 (residues 42–56) and helix h14 (residues 488–494) of Prc involved in interacting with NlpI can be well defined and almost identical between the structures of Prc$^{SK}$-NlpI-MepS and NlpI-Prc-K477A (Supplementary Fig. 18A). Unfortunately, the ligand-binding PDZ domain of Prc$^{SK}$ is poorly defined in the density map of the Prc$^{SK}$-NlpI-MepS complex (Supplementary Fig. 18B), suggesting that the ligand-binding PDZ domain is highly dynamic. Based on the previous structural studies[45], helix h9 (residues 231–242) of the activated Prc-K477A is relatively rigid, while that of the resting Prc-S452I/L252Y becomes flexible and partially unfolded with poor quality of electron density. The helix h9 of Prc can be clearly observed in our Prc$^{SK}$-NlpI-mMepS and can be well superimposed with that of Prc-K477A in complex with NlpI (Supplementary Fig. 18C), confirming that Prc$^{SK}$ adopts the active conformation in the Prc$^{SK}$-NlpI-MepS complex. Moreover, in each Prc$^{SK}$, we observed a co-purified peptide at the proteolytic site, with the unidentified peptide represented by a poly-Ala model (Supplementary Fig. 19). The locations and orientations of peptide fragments show a high similarity to those in the structure of the NlpI-Prc-K477A complex[43]. Owing to the missing electron density of the ligand-binding PDZ domain, neither the position of PDZ domain nor the distance between the C-terminus of mMepS and the PDZ domain can be clearly determined. However, our complex structure still revealed that the C-terminus of MepS2 faces the concave surface formed by the h2-h6 of NlpI (Supplementary Fig. 20), while that of mMepS-1 is solvent-exposed (Supplementary Fig. 19), suggesting that Prc might first target mMepS-1 for degradation rather than mMepS-2 according to the localization of their C-terminal tails.

Furthermore, we used gel-based assays to examine whether the N-terminal residues of mMepS affect the degradation efficiency of the Prc-NlpI system, as it plays a role in interacting with NlpI. The truncated dN36-MepS mutant and F31A variant prominently abolished the enthalpy-driven interaction with NlpI, resulting in no favorable enthalpy changes in the ITC experiments (Fig. 4). Consistent with previous reports[42], in the presence of NlpI, mMepS was rapidly degraded by Prc, while the degradation of dN36-MepS by the NlpI-Prc system was less efficient (Fig. 5C and Supplementary Fig. 21A). MepS-F31A was also found to dramatically hinder the efficient MepS degradation by the Prc-NlpI complex (Fig. 5C and Supplementary Fig. 21B), in line with its weak interaction with NlpI in vitro. Collectively, the altered efficiency of mMepS degradation conferred by the NlpI-mMepS interface mutants further confirms our complex structures and the role of these critical residues in MepS recognition by the Prc-NlpI proteolytic system.

## The N-terminal of MepS largely affects the proteolysis of mMepS in vivo

Previous studies have suggested that in vivo interactions of NlpI with MepS and/or Prc, the Co-Immunoprecipitation assays demonstrated that the MepS interacted with NlpI in the absence and presence of Prc[42]. To examine how the N-terminal of MepS affected the association of NlpI with MepS and/or Prc in vivo, we expressed MepS-N53 using an arabinose-inducible N53 plasmid, encoding a polypeptide lacking residues 54–162. The overexpression of MepS-N53 caused cell morphological changes with the formation of long filaments (Fig. 6A), which was highly similar to Δ*nlpI* cells bearing an increased level of MepS[33,42]. In wild-type *E. coli*, the level of MepS was quite plentiful during the log phase, dropping abruptly in the stationary phase[42]. Therefore, we investigated cellular MepS in the different phases of cell growth by immunoblot analysis of a chromosomal MepS-3XFlag derivative. The result confirmed that MepS-N53 expression dramatically resulted in the accumulation of MepS throughout the exponential and stationary phases (Fig. 6B), while the level of MepS in the non-inducing cells suddenly decreased during the stationary phase. Using the NlpI-specific antibody to investigate the protein level of NlpI, we demonstrated that the NlpI protein was not reduced in the MepS-N53-overexpressing strain (Fig. 6B). Furthermore, we examined how the cell shape changes induced by N53 influenced the cell envelope using

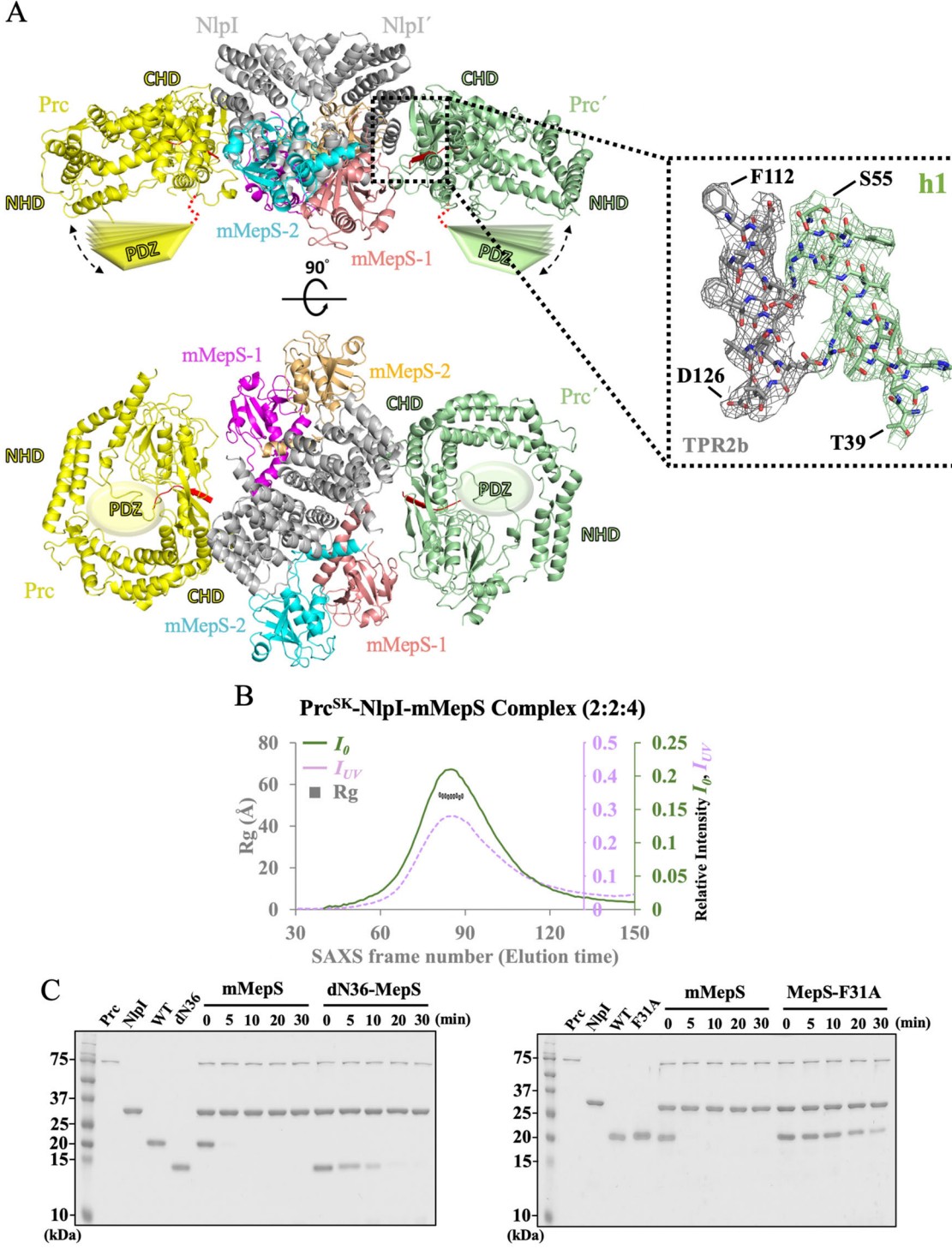

**Fig. 5 | Structural analysis and functional validation of the PrcSK-NlpI-mMepS complex. A** Two orthogonal views of the PrcSK-NlpI-mMepS complex are depicted, highlighting the interaction site between PrcSK and NlpI in cartoon representation. $2F_o$ - $F_c$ maps of TPR2b and h1 from PrcSK are contoured at 1.0 σ as grey and pale green mesh, with the first and last residues labeled (TPR2b: F112 and D126, h1: T39 and S55). NlpI is represented in gray, PrcSK in yellow and pale green, mMepS1 in salmon and magenta and mMepS2 in cyan and golden. The PrcSK protease forms a bowl-like structure with a lid-shaped PDZ domain connected through a substrate-recognizing hinge. Owing to the flexibility of PDZ domains, cartoon models of PDZ domain are shown to emphasize the high dynamics of this region. The missing PDZ-connected hinge is depicted as a red dotted curve, and the dotted double arrow indicates the motion of PDZ domain. **B** A size-exclusion chromatogram of the PrcSK-NlpI-mMepS complex is presented, featuring the radius of gyration (Rg) as a grey rectangle and zero-angle scattering intensity ($I_O$) as a purple dotted line, plotted along with the absorbance at 280 nm ($I_{UV}$) represented by a green line. The elution time is calculated by multiplying the number of SAXS frames by the frame interval of 2.1 seconds. The exposure time is set at 2 seconds per frame, with a wait time of 0.1 seconds per frame interval. **C** In vitro degradation assays of mMepS and mutant proteins by the Prc-NlpI proteolytic system were conducted. Purified MepS proteins were incubated with Prc in the company of NlpI at 37 °C. Protein samples were collected at the indicated time points and analyzed using SDS-PAGE gels followed by Coomassie blue staining. The residual levels of substrate were determined by quantifying the protein band intensity using ImageJ. The data represent a single experiment out of three independent (n = 3) measurements. Source data are provided as a Source Data file.

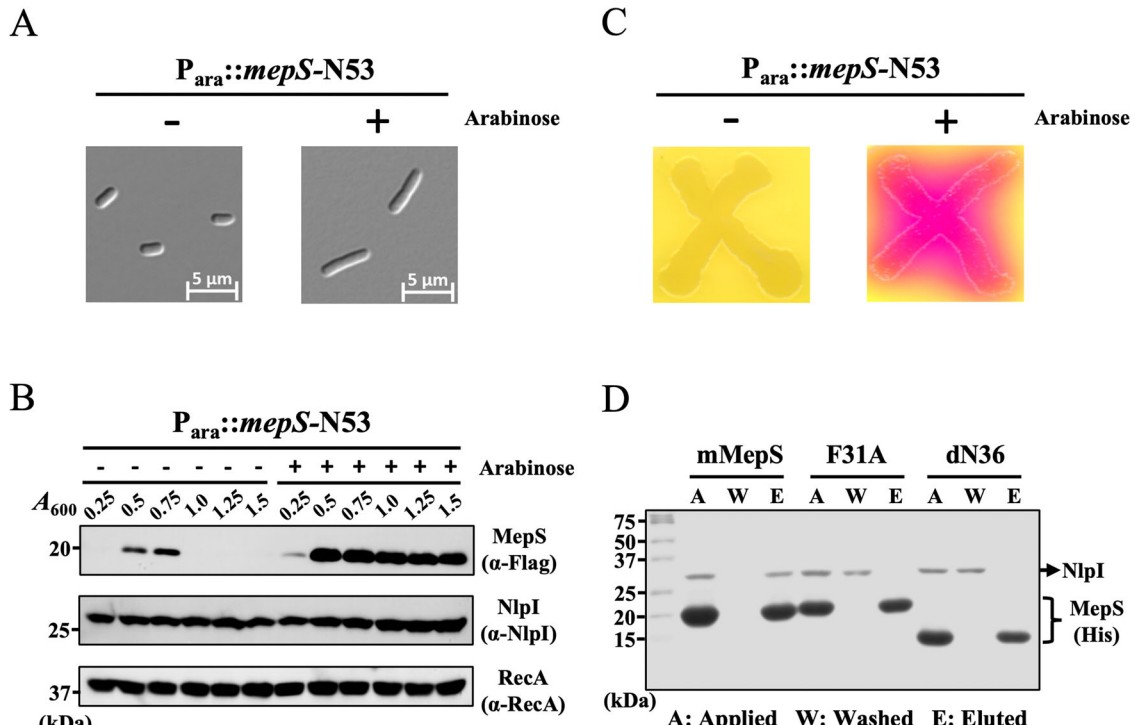

**Fig. 6 | The overexpression of the N-terminal region of MepS has physiological effects on the morphology and cell wall integrity of *E. coli*. A** Cell morphology was visualized using differential interference-contrast (DIC) microscopy to examine the impact of MepS-N53 overexpression. MG1655 WT cells carrying a plasmid with meps-N53 were cultured in LB medium at 37 °C with or without arabinose induction. The overexpression of MepS-N53 resulted in noticeable changes in the morphology of E. coli. Data from one representative experiment were obtained using $n = 3$ biologically independent samples, with scale bars indicating 5 μm. **B** Cellular levels of MepS were evaluated at different growth stages using immunoblot analysis of a chromosomal MepS-3XFlag derivative. Overexpression of MepS-N53 led to a significant increase in MepS levels during both the exponential and stationary phases, whereas MepS levels in non-inducing cells sharply decreased during the stationary phase. Additionally, using an NlpI-specific antibody, it was confirmed that NlpI protein levels were confirmed remained unchanged in the MepS-N53 overexpressing strain. RecA was used as a loading control in the analysis. Data from one representative experiment were obtained using $n = 3$ biologically independent samples. **C** The impact of N53-induced cell shape changes on the cell envelope was assessed using an envelope integrity assay with CPRG, a galactopyranosidase substrate indicating envelope defects. Cells of MG1655 wild-type strain, harboring a plasmid with meps-N53, were cultured on CPRG indicator agar to assess cell wall integrity. CPRG (yellow) is incapable of permeating intact Gram-negative envelopes. The transformation of CPRG to CPR (red) through intracellular β-galactosidase activity serves as an indicator of compromised envelope integrity. Data of one representative experiment were obtained using $n = 3$ biologically independent samples. **D** Pull down assay was performed to demonstrate the interaction between NlpI and mMepS, as well as MepS mutants. The assays were conducted with physiological concentrations of NlpI and MepS at 2 μM and 19 μM, respectively. The samples from $Ni^{2+}$-NTA pull-down assays of mMepS-His, F31A-His, and dN36-His with NlpI were subjected to SDS-PAGE analysis, including the applied (A), washed (W), and eluted (E) samples. Data from one representative experiment were obtained using $n = 3$ biologically independent samples.

an envelope integrity assay with a red-β-D-galactopyranoside (CPRG), the substrate of galactopyranoside that serves as an indicator of envelope integrity defects. The strain overexpressing MepS-N53 exhibited compromised envelope integrity (Fig. 6C), suggesting that the cellular MepS level was substantially disturbed by the N-terminal of MepS. We also performed $Ni^{2+}$-NTA pull-down assays to validate the interaction of MepS variants with NlpI at physiological concentrations (19 and 2 μM, respectively), showing that NlpI mainly interacted with wild-type mMepS (Fig. 6D). By contrast, neither the F31A variant nor the dN36 mutant formed a stable complex with NlpI (Fig. 6D). Altogether, these results provide physiological evidence that the N-terminal of MepS impacts the morphogenesis of *E. coli* by altering the interaction between NlpI and MepS.

## Discussion

This work provides structural insights into how NlpI facilitates MepS colocalization, thus enhancing peptidoglycan hydrolysis and subsequent degradation by the tail-specific protease Prc. During the exponential growth phase, the role of NlpI is to recruit MepS and bring about the avidity effect toward PG binding. This suggests that the MepS-NlpI interaction may regulate cellular processes involving these proteins, rather than simply serving as a waiting area for protease-mediated degradation. Furthermore, the binding of MepS to NlpI becomes weaker in the presence of Prc. Consequently, MepS could move or rotate within the cradle formed by NlpI and Prc at the stationary phase, allowing its flexible C-terminus to reorient toward the PDZ domain of Prc. The PDZ domain of Prc exhibits low specificity in recognizing the C-terminal tails of substrates[52], suggesting that the dynamic PDZ-ligand interactions benefit the translocation process for further proteolysis. The PDZ domain of Prc likely switches between resting and activated states, suggesting its inherent dynamism. Due to this dynamic nature, direct observations of interactions between the C-terminus of MepS and the PDZ domain of Prc may not be straightforward.

We also found that deletion of the intrinsically disordered N-terminal of MepS seriously impacts the intermolecular interactions between NlpI and MepS, as detected by NMR and ITC experiments, indicating that the N-terminal of MepS takes part in NlpI binding. In addition to the deletion of the first 36 residues of MepS (dN36-MepS), the point substitutions, L24R and F31A, in the N-terminal of MepS show that the NlpI-MepS interaction is prominently contributed by the critical residues with high hydrophobicity, especially when a mutation at the aromatic side-chain of residue F31 to the moderately hydrophobic Ala almost cancels out the measurable enthalpy changes and impairs

the folding-upon-binding event. Contrarily, in the case of the conformational selection mechanism[53], the nonpolar amino acid Ala is supposed to have a greater tendency toward α-helix formation than Phe[54,55]; a substitution of Phe to Ala may increase the propensity of the helical structure and then facilitate the interaction. Given that Ala substitution hardly hampers the helix formation in most cases, the mutation effect of our F31A mutation is expected to correlate directly with the decreased hydrophobicity of this key residue, thus weakening the interaction with NlpI and the subsequent folding.

We examined the conservation of the MepS and NlpI structures, respectively, through the sequence alignments of the similar sequences identified through BLAST search (Supplementary Figs. 22, 23), suggesting that the homologs of MepS and NlpI both share a common tertiary and quaternary organization. Based on the Protein DisOrder prediction system[56], the N-terminals of the MepS homologs contain intrinsically disordered regions. Analysis of the N-terminal sequence of MepS showed conservation of the hydrophobic residues L24 and F31, which are involved in the interaction with NlpI (Supplementary Fig. 22), implicating that the *E. coli* MepS-NlpI interaction may occur and form similar complexes in other Gram-negative species. Therefore, the complex structures potentially inform important aspects of cell envelope biology in Gram-negative bacteria.

This study aimed at gaining structural insight into how the outer membrane-anchored lipoprotein NlpI recruits the endopeptidase MepS during the exponential growth phase and then contributes to regulating the cellular MepS level by associating with the tail-specific protease Prc to target MepS for degradation at the stationary stage[39,40]. Our data provide the structure of NlpI in complex with endopeptidase MepS determined at 2.8 Å resolution, revealing that a homodimeric NlpI symmetrically interacts with two asymmetric MepS dimers via a folding-upon-binding event. The NlpI-enabled mMepS colocalization may bring about the avidity effect toward PG binding (Supplementary Fig. 24), suggesting that once one MepS recognizes and binds to a target, other MepS molecules then join to cleave the D-Ala-*meso*-DAP crosslink. Thus, this complex structure could explain why the addition of NlpI enhances the activity of MepS against muropeptides, as described in a previous study[39]. Based on the current results, we propose a simplified model of how NlpI may assist with coordinating the PG multi-enzyme complex, including LpoA (a synthesis regulator), PBP1A (a bifunctional synthase), NlpI and several endopeptidases during the log growth phase (Fig. 7A). Furthermore, the MepS level is known to be degraded by the tail-specific protease Prc in complex with the adaptor NlpI, as Prc alone hardly cleaves MepS. The abundance of MepS was observed in the log of the growth curve, while the level of MepS greatly decreased at the beginning of the stationary phase. Therefore, we present a cartoon model of how NlpI acts as a bridge between MepS and Prc, therefore promoting the efficient degradation of MepS by Prc in the stationary phase (Fig. 7B). Overall, this study advances our knowledge of how NlpI recruits multiple MepS molecules and reveals the essential structural details required to understand the mechanism of NlpI-enabled MepS colocalization. Moreover, the mechanism of NlpI-dependent proteolysis is clarified by the structure of the Prc-NlpI-mMepS complex, providing the structural evidence that adaptor NlpI brings MepS and the protease Prc together for efficiently degrading MepS by Prc.

## Methods

### Cloning, mutagenesis, protein expression and purification
We used an *E. coli* expression system for producing all recombinant proteins in this study. All recombinant proteins with a 6xHis tag were purified by immobilized metal affinity chromatography and SEC. The DNA sequence encoding mature MepS (residues 1–162, corresponding to residues 28–188 in the MepS precursor) was cloned into either pET21a or pET28a vector, which expressed a C-terminal His-tag or an N-terminal His-tag and a TEV cleavage site, respectively. The lipidation of MepS, which is located at residue C1 in the mature form of MepS[46], has been replaced with a Met residue. The wild-type gene of NlpI (20–294 residues) was cloned into pET28a vector with an N-terminal His-tag and a TEV cleavage site, and purified mature NlpI was soluble up to 200 mg/ml in in 10 mM Tris/HCl at pH 8.0 and 10 mM NaCl. The lipidation of NlpI, occurring at residue C19, is not included in our construct[36]. The NlpI-ΔN (37–294) mutant, devoid of residues 1–36, was cloned into a pET28a vector containing an N-terminal His-tag[43]. Prc (1–682 residues) was cloned into pTrc99A vector with a C-terminal His-tag as described previously[57]. All MepS and Prc variants were obtained by using site-directed mutagenesis (QuikChange™ site-directed mutagenesis kit from Stratagene). All plasmids harboring gene of interest were transformed into *E. coli* BL21 (DE3) cells, which were grown in 1.5 L LB with the corresponding antibiotics to $OD_{600}$ 0.6–0.8 at 37 °C. The overexpression of target proteins was induced with 0.5 mM isopropryl-thiol-β-D-galactoside (IPTG) at 25 °C overnight. Cells were harvested by centrifugation and disrupted using a French press (Avestin) in buffer containing 500 mM NaCl, 25 mM NaPi, 5% glycerol, and 5 mM β-mercaptoethanol, pH 8.0. The lysate was centrifuged at 50000 x *g* for 1 hr at 4 °C and the supernatant was applied to nickel-nitrilotriacetic acid resins (Ni-NTA, Qiagen) and the protein sample was collected and concentrated to approximately 5 ml using Amicon (Millipore) or Vivaspin (Satorius) centrifugal concentrators at 3200 x *g* at 4 °C. The sample was subsequently applied to HiLoad 16/600 Superdex 200 or Superdex 75 column (GE Healthcare) pre-equilibrated with 20 mM MES, 300 mM KCl, and 2 mM β-mercaptoethanol, pH 6. Except for MepS proteins, NlpI and Prc proteins were further purified by MonoQ 5/50 GL column (GE Healthcare) pre-equilibrated with 20 mM Tris-HCl, and 50 mM NaCl, pH 8.0. For NMR experiments, isotopically labeled proteins were prepared in *E. coli* BL21 (DE3) cells grown in minimal (M9) medium either with $^{15}NH_4Cl$ or with $^{15}NH_4Cl$ and $^{13}C$-glucose for uniform labeling of isotope(s). Protein purity was analyzed using SDS-PAGE and fractions with highest protein purity were collected.

### NMR spectroscopy
All NMR data were collected on Bruker 800-MHz spectrometers (Bruker Biospin, Karlsruhe, Germany) at 298 K. For resonance assignment of full-length MepS, the following experiments were conducted: 2D $^{15}N$-$^{1}H$ HSQC, 3D HNCACB, 3D HN(CO)CACB, 3D HNCA, 3D HN(CO)CA, 3D HNCACO, and 3D HNCO. The NMR spectra were acquired using 600 μM $^{15}N$, $^{13}C$-labeled protein in NMR buffer containing 0.2 M Na/K phosphate, pH 7.0 with 5% $D_2O$. $^{1}H$-$^{15}N$ NOE (heteronuclear NOE) data were recorded in an interleaved fashion with alternating saturated and unsaturated transients and a recycle delay of 4 s was used. All NMR spectra were processed with NMRPipe[58] and analyzed using NMRView.

For NMR titration experiments, $^{1}H$-$^{15}N$ HSQC-TROSY spectra were acquired at 298 K for each sample, and the intensity ratios ($I/I_O$) were calculated, where $I$ and $I_O$ correspond to the peak intensity of the bound and free samples, respectively. Our titration results revealed a lack of CSPs and consistently showed a reduction in cross-peak intensity upon the addition of NlpI, suggesting the formation of a substantial complex. Therefore, we have included the mean values for the NMR titration and highlighted the residues exhibiting intensity ratios ($I/I_O$) higher than the average plus one SD, attributable to the point mutations.

1D $^{15}N$-edited relaxation experiments were performed to measure average $^{15}N$ $T_1$ and $T_2$ relaxation times for 99%-$^{15}N$ mMepS (residues 2–162) and dN36-MepS (residues 37–162). The data were acquired on a

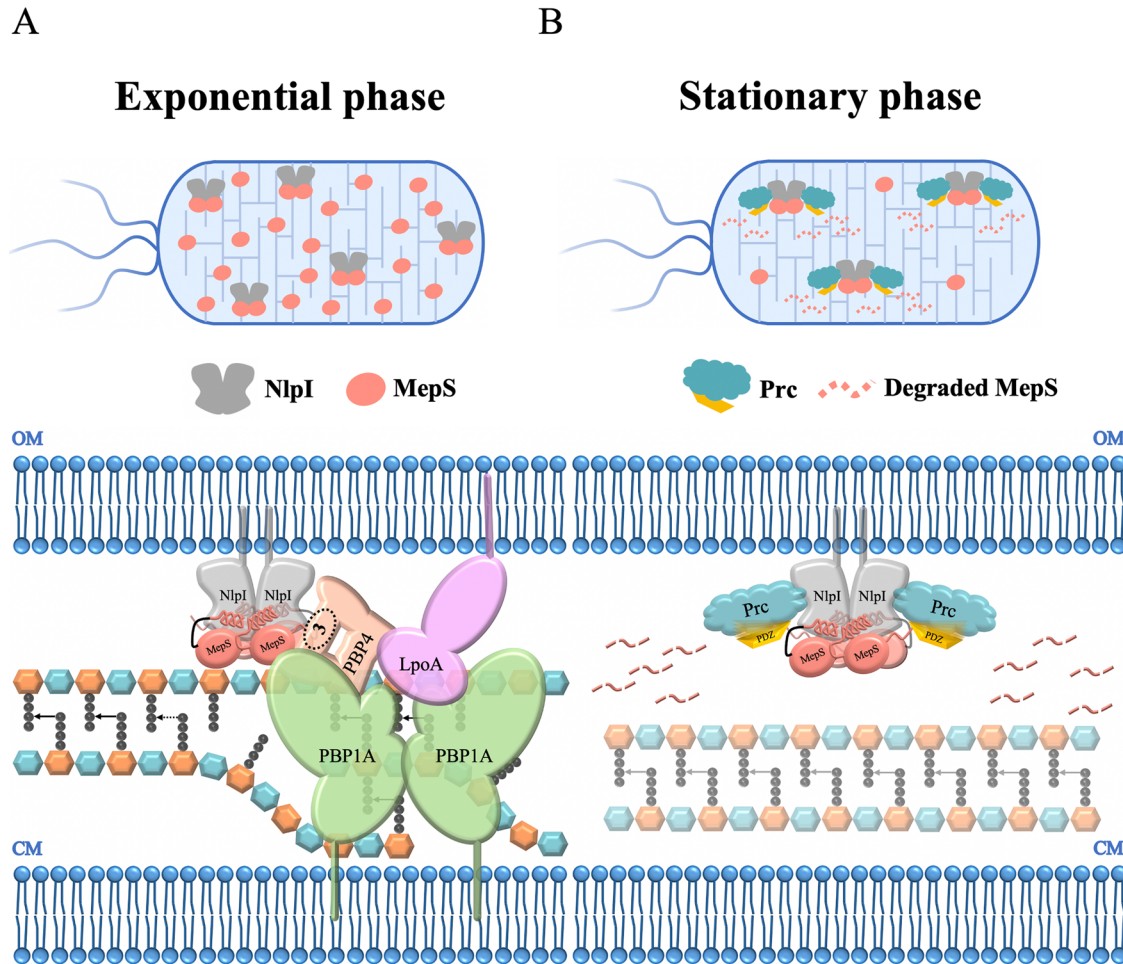

**Fig. 7 | Model illustrating the scaffolding function of NlpI for coordinating PG endopeptidases and PG synthetic machinery. A** A simplified model for the potential role of NlpI in coordinating the PG multi-enzyme complex, involving LpoA (a synthesis regulator), PBP1A (a bifunctional synthase), NlpI, and various endopeptidases during the exponential growth phase. This model involves the outer membrane (OM) lipoprotein NlpI, which acts as an adaptor protein to scaffold different endopeptidases into a trimeric complex. Specifically, NlpI forms a complex with MepS and PBP4, and potentially with PBP7[39]. Afterwards, the PBP4 (or PBP7), as a linking partner, could further recruit the PG synthesis machinery, PBP1A/LpoA to facilitate the formation of PG multi-enzyme complexes (PBP1A·MepS·NlpI·PBP4·LpoA or PBP1A·MepS·NlpI·PBP7·LpoA). PBP4 is found to interact with NlpI through its domain 3, as encircled and labeled in the model. Some stem peptides are removed for clarity. This proposed model provides insights into the potential mechanisms by which NlpI coordinates the activities of various enzymes involved in peptidoglycan synthesis. **B** The proposed model illustrates the direct control of the protein levels of endopeptidase MepS by the Prc-NlpI proteolytic system, which is crucial for maintaining cell envelope integrity during the stationary phase. As bacterial cells transition into the stationary phase, MepS-bound NlpI recruits the *E. coli* periplasmic PDZ-protease Prc, leading to rapid degradation of MepS. This regulatory mechanism plays a vital role in ensuring cell wall stability under stationary phase conditions.

Bruker 800 MHz spectrometer at 298 K using pseudo-2D $^{15}$N $T_1$ and $T_2$ gradient experiments. $T_1$ spectra were acquired with delays, $T = 20, 50, 100, 200, 300, 400, 600, 800, 1000, 1200,$ and 1500 ms, and a relaxation delay of 3 s. $T_2$ spectra were acquired with CPMG delays, $T = 16, 32, 48, 64, 80, 96, 128, 160, 192, 240,$ and 320 ms, and with a relaxation delay of 1.5 s. To minimize contributions from the disordered regions of mMepS, $^{15}$N $T_1$ and $T_2$ values were extracted by plotting the decay of integrated $^1$H$^N$ intensity between 9.0 and 9.5 ppm and fitting the curves with standard exponential equations using the program 't1guide' within Topspin 4.0.6 (Bruker BioSpin). The $\tau_c$ was calculated from the $^{15}$N $T_1/T_2$ ratio using the following approximation of the literature relaxation equation:

$$\tau_c = \sqrt{\frac{6T_1}{T_2} - 7/4\pi\upsilon_N}$$

where $\upsilon_N$ is the resonance frequency of $^{15}$N in Hz.

### Size exclusion chromatography (SEC) analysis
Purified mMepS and dN36-MepS were mixed at a molar ratio of 1:1, and incubated with 1.5-fold protein amount of NlpI to a volume around 500 µl. The protein mixtures were centrifuged to remove aggregates, applied onto Superdex200 Increase 10/300 GL column with 500-µl loop, and analyzed at a flow rate of 0.3 ml/min using an AKTA purifier system. Luted proteins were detected by UV absorbance at 280 nm. The elution profiles were compared to injections of a mixture of mMepS and dN36-MepS or NlpI protein alone for data analysis. The peak fractions, collected by the AKTA fraction collector, were examined by SDS-PAGE using Coomassie blue staining.

### Crystallization and data collection
To determine the initial crystallization conditions, both of the concentrated NlpI·mMepS (22.2–26.7 mg/ml) and Prc$^{SK}$·NlpI·mMepS (24.2–36.3 mg/ml) proteins were screened automatically against 960 conditions from crystal screen kits (Hampton Research, USA;

Molecular Dimensions, UK) by the sitting-drop vapor-diffusion method at 277 K. The NlpI-mMepS and the Prc$^{SK}$-NlpI-mMepS proteins were first crystallized by using Structure Screen 1 No. 12 (1.4 M sodium acetate, 0.1 M sodium cacodylate, pH 6.5) and Wizard III B1 (8% PEG 4000, 1 M sodium acetate, pH 4.6), respectively, as the reservoir solution. Better crystals of the NlpI-mMepS complex were obtained by optimizing the reservoir composition (*e.g.*, pH and precipitant concentration) to 1.3 M sodium acetate, 0.1 M sodium phosphate monohydrate, and 0.1 M sodium cacodylate, pH 6.3. Similarly, the better crystals of the Prc$^{SK}$-NlpI-mMepS complex were produced by slightly changing the reservoir composition to 6%-10% PEG4000 and 1 M sodium acetate, pH 4.6. All optimized crystals were prepared manually using the hanging-drop vapor-diffusion method at 277 K and reached a suitable size for X-ray diffraction in 14 days. Crystals containing the protein complexes were confirmed by washing crystals three times with artificial mother liquor and performing SDS-PAGE analysis on the washed crystals, using the last wash step as a negative control. The crystals were cryoprotected by transferring them into well solutions containing 20–25% ethylene glycol before flash-freezing in liquid nitrogen for data collection. The diffraction data were collected at the beamlines TPS 05 A and TPS 07 A of the National Synchrotron Radiation Research Center (Taiwan). All the datasets were processed using the HKL2000 package[59].

### Structure determination

The crystal structure of the NlpI-mMepS complex was solved by molecular replacement (MR) with the program *Phaser*[60] using the structures of NlpI (PDB ID 1XNF) and MepS (PDB ID 2K1G) as the search models. The NlpI-mMepS crystal belongs to space group C2 and contains twelve molecules, including four NlpI molecules and eight mMepS molecules, in an asymmetric unit. The model building and structure refinement were completed by the program *Coot*[61] and *Refmac*[62]. The crystal structure of the Prc$^{SK}$-NlpI-mMepS complex was determined by MR using the program *Phaser-MR* in the CCP4 suite[63]. The deposited NlpI-Prc-K477A complex (PDB ID 5WQL) and the refined mMepS from the NlpI-mMepS complex solved in this study were used as a composite starting model. The Prc$^{sk}$-NlpI-mMepS crystal (space group P4$_3$ 2$_1$ 2) contained eight molecules, including two Prc$^{SK}$, two NlpI molecules, and four mMepS molecules, in an asymmetric unit. The $2F_o$-$F_c$ difference Fourier map showed clear electron densities for most amino acid residues of NlpI and mMepS, but some regions from 133 to 148, 185 to 192, 272 to 285, 309 to 333, 345 to 364, 384 to 390, 526 to 533, 558 to 563, 638 to 644 (C chain of the Prc$^{SK}$ structure) and 181 to 194, 279 to 283, 315 to 334, 352 to 362, 384 to 389, 431 to 436, and 471 to 477 (D chain of the Prc$^{SK}$ structure) lacked electron densities, presumably due to disorder or conformational changes. Subsequent refinement with the incorporation of water molecules was performed according to the 1.0 σ map level. During the refinement, however, the crystallographic $R_{work}$ (0.282) and $R_{free}$ (0.366) have maintained higher values and large gaps. To improve the issue, two better datasets were merged into one with higher quality to reduce $R_{work}$/$R_{free}$ values (0.239/0.281). The model building and the structural refinements were performed by the program *Coot*[61] and *Phenix*[64] with translation libration screw (TLS) and non-crystallographic symmetry (NCS) restraints, respectively. All structures were validated by MolProbity (http://molprobity.biochem.duke.edu/). Data collection and structure refinement statistics are summarized in Supplementary Table 1. All figures were generated using PyMOL (Version 2.5.2, Schrödinger, LLC). The atomic coordinates and structure factors of the NlpI-mMepS and Prc$^{SK}$-NlpI-mMepS complex structures have been deposited at the Protein Data Bank with the codes 8XUP and 8XUD, respectively.

### Isothermal titration calorimetry (ITC)

ITC titrations were performed on an ITC200 calorimeter system (MicroCal Inc.) at 20 °C in ITC buffer containing 20 mM HEPES, pH 7.5,

300 mM NaCl and 2 mM β-mercaptoethanol. Protein samples in both cell and syringe were buffer-exchanged in the same ITC buffer by 7 K MWCO, Zeba™ Spin Desalting Column (Thermo Scientific) prior to all titrations. All MepS proteins were diluted to 30 μM, and NlpI was prepared at final concentrations of 300 or 600 μM. For each ITC experiment, 40 μl NlpI solution was injected into the sample cell containing 200 μl MepS proteins by 16 consecutive injections with 0.5 μl for the first and 2.49 μl for the rest. Control titrations were conducted, which involved the titration of ligands into buffer or titration of buffer into protein solutions, as illustrated in Supplementary Fig. 25. The data analysis was conducted using one set of sites fitting model incorporated in Origin 7.0 (MicroCal Inc.).

### SAXS measurements and analysis

SEC-SAXS experiments were performed at the Taiwan Photon Source (TPS) 13A BioSAXS beamline of the National Synchrotron Radiation Research Center, using an integrated system featuring size-exclusion chromatography (Agilent chromatographic system 1260 series) and small-/wide angle scattering (SWAXS), coupled with UV–Vis absorption[65,66]. In these experiments, 100 μL of sample solutions were respectively injected into an SEC column (Superdex 200 Increase 5/150 GL) at a flow rate of 0.2 ml/min. The samples were directed through a quartz capillary (2.0 mm dia.) thermostated at 283 K for simultaneous SWAXS and UV–vis absorption measurements, with orthogonal incidences on the same sample position. With 15 keV X-ray (λ = 0.8266 Å) and sample–detector distances of 2.5 m and 0.3 m, for the two in-vacuum SAXS (Eiger X 9 M) and WAXS (Eiger X 1 M) detectors, respectively, the scattering vector $q$ range covers 0.0065 to 2.6 Å$^{-1}$. Continuous SWAXS data were collected with 2-second per data frame acquisition rate over the elution peak. The frame data of well-overlapped SAXS profiles were averaged, subtracted with scattering from the buffer solution (0.2 M Na/K phosphate) measured before sample elution, and scaled to the absolute intensity in units of cm$^{-1}$ based on the scattering intensity of water at 283 K. The TPS 13A SWAXS data reduction kit (DRK) Ver. 4–88 further processed the WAXS data frames corresponding to those selectively processed SAXS data and merged the SAXS-WAXS data sets into integrated SWAXS data. The quality of the SAXS data was evaluated using ATSAS 3.0.4[67]. Following a previously established methodology[51], the PNM-a complex composition of 2P:2N:4M was determined from the SAXS zero-angle scattering intensity $I_0$ and the sample concentration (determined by UV-vis absorption), as detailed in the supporting information.

### In-vitro degradation assay

The degradation assay was carried out in a degradation buffer containing 20 mM Tris-HCl, pH 8.0, 150 mM NaCl, and 2 mM DTT at 37 °C. Each reaction mixture (20 μl) was composed of 4 μM MepS protein incubated with 1 μM NlpI and 0.5 μM Prc. The protease reaction was stopped by addition of 6x SDS-PAGE loading dye at the indicated time intervals. The collected samples were heated at 95 °C for 10 min and loaded onto 12–15% polyacrylamide gels, followed by Coomassie blue staining. The residual level of MepS protein was determined by the protein band intensity from three independent experiments using ImageJ. The intensity of endopeptidase variants in the 0-min reaction was set to 100%.

### Pull down assay

The pull down assay was carried out in a binding buffer containing 500 mM NaCl, 25 mM NaPi, 5% glycerol, 5 mM β-mercaptoethanol, pH 8.0, at 4 °C. Each reaction mixture (1 ml) consisted of 19 μM His-tagged MepS proteins incubated with 2 μM untagged NlpI in the presence of nickel-nitrilotriacetic acid resins (Ni-NTA, Qiagen). The beads were pre-equilibrated with dH$_2$O and binding buffer. The samples were incubated overnight on a spinning plate at 4 °C before the beads were washed with binding buffer supplemented with 20 mM imidazole. The

retained fraction was eluted from Ni-NTA beads using elution buffer containing 500 mM NaCl, 25 mM NaPi, 400 mM imidazole, 5% glycerol, and 5 mM β-mercaptoethanol, pH 8.0. The collected samples were heated at 95 °C for 10 min and then loaded onto 12–15% polyacrylamide gels, followed by Coomassie blue staining.

## NlpI antisera production

The production of NlpI antisera involved immunizing mice with purified NlpI protein (with the His-tag removed). The overexpression and preliminary purification of NlpI proteins followed the abovementioned protocol. Samples, featuring a TEV protease cutting site between the His tag and NlpI, were incubated with His-tagged TEV protease at a ratio of 1 to 100 w/w in a buffer containing 500 mM NaCl, 25 mM NaPi, 5% glycerol, and 5 mM β-mercaptoethanol, pH 8.0, at 4 °C overnight. The removal of the His-tagged TEV was achieved using a column containing Ni-Nta resins (Qiagen). The cleavage efficacy was confirmed by Coomassie-stained SDS-PAGE. The purification of pure NlpI proteins was accomplished through gel filtration chromatography using a Superdex 200 column (GE Healthcare) and the subsequent production of NlpI antisera was outsourced to Leadgene Biomedical, Inc. (Tainan, Taiwan).

## Western blot for intracellular MepS levels

Strain MR801 (MG1655 mepS-Flag-Kn) was transformed with the plasmid pBAD18 harboring the gene encoding mepS-N53, corresponding to the N-terminal region of mMepS. The resulting transformants were grown in 50 ml LB medium supplemented with ampicillin and kanamycin, with or without 0.02% L-arabinose induction, at 37 °C. Fractions were collected at specified values of O.D.600 values by centrifugation. The harvested cells were suspended and lysed in 1x tris-glycine SDS sample buffer at 95 °C. Next, the samples were resolved using 12% SDS-PAGE and subsequently transferred onto a PVDF membrane preactivated with methanol. The membrane was then blocked with 5% skimmed milk in 1× TBS-T for 1 hour, followed by an overnight incubation with primary antibodies (diluted to 1:100 for α-NlpI, 1:2000 for α-FLAG, and 1:3000 for α-RecA) at 4 °C. RecA served as a loading control. Following three 10-minute washes with 1× TBS-T, the membrane was subsequently probed with secondary antibodies (diluted to 1:5000) conjugated with horseradish peroxidase (HRP) and incubated for 1 hour at room temperature. Prior to signal detection, the membranes were subjected to three additional washes with 1x TBS-T for 10 minutes each and were overlaid with ThermalFisher SuperSignal™ West Atto Ultimate Sensitivity Substrate for detection of HRP.

## Live cell imaging and cell morphology

The MG1655 strains were transformed with the plasmid pBAD18 harboring the genes encoding mepS-N53. The resulting transformants were cultured in 5 ml LB medium supplemented with ampicillin, with or without 0.02% L-arabinose induction at 37 °C until reaching an OD600 of 0.6–0.8. Exponentially growing cells were harvested through centrifugation, resuspended in 1xPBS, and subsequently applied to pads made of 1% agarose in 1xPBS. Imaging was performed using a Zeiss ApoTome.2 microscope equipped with a Zeiss Lens ×100/1.4 Oil DIC objective and a Cascade: ORCA-Flash 4.0 V3 camera (Hamamatsu). The acquired images were recorded and processed using Zen Blue 2.6.

## CPRG assay for assessing cell wall integrity

To evaluate the influence of the MepS-N53 peptide on cell wall stability in E. coli, transformants of MG1655 carrying the pBAD-mepS-N53 plasmid were cultured in 5 ml LB cultures supplemented with ampicillin at 37 °C until reaching an OD600 of 0.6–0.8. Then, bacterial cultures were patch-plated in an 'X' pattern onto LB agar containing 20 μg/ml of CPRG, with or without 0.02% arabinose. The plates were then incubated overnight at 37 °C and subsequently photographed.

CPRG (yellow) is unable to permeate intact Gram-negative envelopes, so the conversion of CPRG (yellow) to CPR (red) serves as an indicator of compromised envelope integrity.

## Reporting summary

Further information on research design is available in the Nature Portfolio Reporting Summary linked to this article.

## Data availability

The atomic coordinates generated in this study have been deposited in the PDB under the accession codes 8XUP and 8XUD. Resonance assignments for the mature MepS generated in this study have been deposited in the BMRB (BioMagResBank) under the accession codes 51949. The protein sequences used in this study were retrieved from UniProt under the accession codes P0AFV4 (mepS), P0AFB1 (nlpI) and P23865 (prc). Previously reported structures retrieved from PDB include 2K1G (MepS), 7V6S (MepS), 1XNF (NlpI) and 5WQL (NlpI-Prc complex). Previously reported assignment retrieved from BMRB: 15603. Source data are provided with this paper.

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

## Acknowledgements

We thank Prof. Jin-Town Wang and Prof. Ching-Hao Teng for their constructive suggestions and for providing plasmids for recombinant protein overexpression. We also thank Prof. Manjula Reddy for the Strain MR801, Dr. Tzu-Ping Ko of ASPC (AS-CF11-113-A11) for constructive discussions about the crystal structures, and Dr. Shu-Chuan Jao in the Biophysics Core Facility, funded by Academia Sinica Core Facility and Innovative Instrument Project (AS-CFII-111-201), for support with ITC data analysis. We acknowledge the National Science and Technology Council (Taiwan) for research support (NSTC 112-2740-M-002-006, NMR004900); the Imaging Core at the First Core Labs, National Taiwan University College of Medicine (NTUCM), for the technical support in image acquisition and analysis; the services provided by the Protein X-ray Crystallography Facility at the Institute of Biological Chemistry, Academia Sinica, Taiwan. We are grateful to the staff of the Biomedical Resource Core at the First Core Labs, NTUCM, for their technical assistance. We deeply appreciate the antibody service provided by Leadgene Biomedical, Inc. (Tainan, Taiwan) for the development and purification of the NlpI anitobody. Portions of this research were carried out at beamlines 13C1, 13B1, 13A1, 15A1, 05A and 07A of the National Synchrotron Radiation Research Center (Taiwan). This work was supported by funds from National Taiwan University (under Grants NTUJP-111L7225 and 112L7227 to S.-R.T.) and the Ministry of Science and Technology, Taiwan (MOST 109-2113-M-002-018 and NSTC 112-2113-M-002-016 to S.-R.T.).

## Author contributions

C.I.C. and S.R.T. conceived and designed the study. S.W., H.C.T., and S.J.H. performed NMR experiments. S.W. and Y.Y.C. performed ITC titrations. S.W., C.H.H. and T.S.L. performed the structure calculations and validation. S.W., Y.Q.Y. and U.S.J. performed and analyzed the SAXS experiments. S.W., Y.S.F., and S.W.W. performed and analyzed the in-vivo and pull-down experiments. S.W., Y.Q.Y., C.H.H., and S.R.T. wrote the manuscript. All authors commented and contributed to the approved final version of the manuscript.

## Competing interests

The authors declare no competing interests.
