## [Peer Review File · Nature Communications]

Reviewers' Comments:

Reviewer #1:

Remarks to the Author:

The structural and biochemical study by Shen Wang and company presents the structure of the NlpI adaptor complexed with the endopeptidase MepS, revealing how NlpI recruits and influences multiple MepS molecules, promoting PG expansion. NlpI binding induces a structural change in the intrinsically disordered N-terminal of MepS, promoting MepS dimerization and enhanced activity in PG hydrolysis. The protein level of MepS binds to NlpI. The structure of the Prc-NlpI-MepS complex demonstrates that NlpI facilitates the interaction between MepS and Prc, leading to efficient degradation of MepS by Prc. The findings shed light on how NlpI enhances the binding efficiency of cellular endopeptidases and directs MepS degradation by Prc. Please see my comments below.

Major Concerns

1. Understanding the significance of the study and the advancements it made in the context of existing knowledge was difficult to grasp on a global scale based solely on this paper.

- Suggestion: The introduction should be expanded to provide more context and clearly state the study's contributions and how they advance the current knowledge in the field.

2. The study focuses on the binding interactions between NlpI, PrcSK, and mMepS mutants but does not explore the broader context of other potential interacting partners mentioned in lines 78-92 and again 329-330.

- Suggestion: Acknowledge the limited scope and suggest avenues for future research to investigate other potential interacting partners and their relevance.

3. The study is mainly based on in vitro experiments, lacking in vivo validation of mutants, etc., to understand and demonstrate the relevance of the findings in a more physiological setting.

4. The extent of the proposed model's applicability to other bacterial systems remains uncertain or unclear from the work presented.

- Suggestion: Discuss the limitations of the proposed model and consider how it might be tested in other bacterial systems to establish its broader applicability.

5. The authors should have included controls where the individual proteins (NlpI, PrcSK, and mMepS) are tested independently to confirm their binding and interaction, and also test the mutants in isolation to verify their effects on binding.

6. The crystal structures (3.5 Å for PrcSK-NlpI-mMepS and 2.8 Å for NlpI:mMepS complex) do not fully support all the conclusions drawn in the study (see below)

7. The electron density of the PrcSK-NlpI-mMepS complex in Figure SA is not convincing and does not clearly show the complex at the defined resolution. The maps are barely interpretable.

8. The electron density map for Figure S16 is also not convincing.

9. The quality of chains A through L in the PDB validation for the Crystal structure of adaptor NlpI in complex with endopeptidase MepS and PDZ-protease Prc is poor for the reported resolution.

Minor Concerns

1. Lines 116-117: "The intrinsically disordered N-terminal of endopeptidase MepS is crucial for the interaction with adaptor NlpI." appears to be an overstatement.

2. In lines 119-120: "The NlpI-interacting EPase MepS has been reported to adopt a papain-like $\alpha+\beta$ fold with a novel catalytic triad comprising C68, H119, and H13142,44 . This sentence lacks clarity and precision. The information provided is not clear in explaining how the "papain-like $\alpha+\beta$ fold" and the "novel catalytic triad" relate to the interaction with NlpI.

3. Lines 123-127: "Based on the reported NMR structure of MepS42, truncated mutant dN36-MepS (residues 37-162) was selected for our initial titration experiments (Figure S1), while the first 36 N-terminal residues, which were previously suggested to be significantly disordered in the screening by NMR42, was suggested to be dispensable for a catalytic mechanism in the cysteine peptidase family." The sentence structure is awkward and may lead to confusion. It is not clear what the authors are trying to convey regarding the relevance of the N-terminal residues in the catalytic mechanism.

4. Lines 131-133: "Interestingly, the NMR spectra of dN36-MepS in the absence and presence of

unlabeled NlpI at a molar ratio of 1:1 (Fig. 1B) were very similar, suggesting that NlpI scarcely binds to dN36-MepS." The authors state that NlpI scarcely binds to dN36-MepS based on the similarity of NMR spectra in the absence and presence of NlpI. However, this conclusion lacks a clear quantitative analysis or statistical evidence to support it in the text.

5. Lines 135-143: "We next found that NlpI prominently induced dramatic variations in the 2D spectral features of mMepS and the linewidth of the bound-state resonances was hardly observed upon NlpI binding (Fig. 1C)." This statement was unclear and needs additional details or clarification regarding the nature of the "dramatic variations" and "linewidth of the bound-state resonances."

7. Lines 213-219: "Compared with WT mMepS, the truncated dN36-MepS mutants showed no heat changes with the titration of NlpI, suggesting that the first 36 N-terminal residues prominently contribute to NlpI binding (Fig. 4B and Table S2)." The statement suggests that the first 36 N-terminal residues "prominently" contribute to NlpI binding. The phrase "no heat changes" needs further explanation and clarification.

8. Lines 260-262: "To understand how NlpI enhances the efficiency of mMepS proteolysis by Prc, we determined the crystal structure of PrcSK-mMepS-NlpI at 3.5 Å, revealing that two monomeric PrcSK bind to hetero-hexameric NlpI-mMepS to form a 2:2:4 hetero-octameric complex (Fig. 5A)." The authors state that the complex formed is a "2:2:4 hetero-octameric complex" may need more evidence or justification for this particular stoichiometry. More details and analysis might be needed.

Reviewer #2:

Remarks to the Author:

Wang and colleagues present two fabulous new crystal structures of the outer membrane lipoprotein NlpI bound to MepS and protease Prc. The structures are interesting and potentially inform on important aspects of cell envelope biology in *E. coli*.

The co-structure of the NlpI/MepS/Prc is particularly impressive and will certainly help the field to understand the interactions of NlpI. The complex also defines a structural basis on which to build further understanding of MepS endopeptidase degradation in the future.

There is some very nice supporting data that documents the importance of the MepS N-terminus in the interaction with NlpI. The NMR experiments support a disorder-to-order transition in the MepS N-terminus during NlpI binding. The use of multiple techniques to further investigate the NlpI/MepS interaction is commendable.

However, there are some concerns over the supporting biochemical data (and interpretation) that should be addressed. In particular, the SEC, SPR and ITC show inconsistencies that need to be resolved. Fortunately, these data most likely can be clarified by appropriate repeats and inclusion of missing controls.

Some comments below.

1. For the size exclusion data, the authors perform an NlpI-binding experiment in competition between full length MepS (mMepS) and MepS lacking its N-terminus (dN36MepS). While the competition experiment is welcome, the various simpler controls should also be performed and reported. For example, there should be runs of each individual protein (mMepS alone, dN36MepS alone, NlpI alone) as controls, and non-competitive pairwise binding experiments - ie mMepS+NlpI, and dN36MepS+NlpI. It will be particularly interesting to see whether MepS lacking its N-terminus (dN36MepS) can still bind NlpI (absent of competition with the wild type). These experiments should also help resolve the disparity between the SPR and ITC results (see below).

2. The ITC data follows up on the structural observations very well. The authors identify a number

of interface residues from the structure(s) and then measure pairwise interactions between MepS and NlpI for each mutation. The data shows that the MepS N-terminus is especially important for binding. The data also suggests that the F31A mutation completely breaks the interaction - which is consistent with F31 being located in the MepS N-terminus.

However, no repeats are reported - so these are presumably single ITC measurements, and the main text overanalyses the very small differences between K_d values measured. (Using Table S2, the measured K_d values are: 0.2 μM, 1 μM, 3 μM, 6 μM, 2 μM, and 0.3 μM.)

The K_d values in the text are reported with tremendous precision (ie two decimal places on the μM measurements - ie 10 nM precision). The results are then discussed as if this level of accuracy were real.

Clearly, the N-terminal deletion and F31A mutant affect binding - but the other K_d values are probably not far outside experimental error. L24R, Q28A and D39A are all very similar measurements - and there is no justification to draw any insight from closely comparing these values.

The paper would be better served if it discussed the K_d measurements as minimal changes to binding affinity for all mutants except F31A and the N-term deletion. It should also report the measured K_d values in Table S2 (not just K_a) and only give values to the nearest μM in the text.

There should also be at least one further repeat for mMepS (the WT), dN36MepS, and MepS-F31A, before these results are considered reproducible. In fact, if the WT were to be measured three times, it would give a much more realistic impression of experimental variability for this method and better inform the meaning of the small differences in K_d observed between WT and most mutants.

3. The Surface Plasmon Resonance (SPR) data is not consistent with ITC results - even though it is reported as if the two are in step.

The SPR gives a much higher affinity than ITC for NlpI and the WT MepS (mMepS) at 0.03 μM, but also suggests that the MepS lacking the N-terminus, or incorporating the interface-breaking F31A mutant, still have K_d values ~4.75 μM.

A K_d value of 5 μM is similar to (or even slightly tighter than) the affinity between EDTA and calcium. Thus SPR suggests the mutants are still forming a tight complex with NlpI.

The K_d values reported by SPR are therefore very different to the ITC. The ITC data implies F31A and MepS lacking the N-terminus are essentially non-binding - while the SPR suggests quite a tight interaction (K_d<5μM).

The ITC should be repeated to check that the non-binding results are correct, and further controls should be performed for the SPR to ensure these are not measuring something other than the protein:protein interaction. Reporting the SEC for NlpI/MepS, NlpI/dN36MepS and NlpI/MepS-F31A would also be extremely useful to check whether the ITC or SPR is more likely to be correct.

4. Finally for the discussion, the architecture of the NlpI/MepS/Prc complex seems to orient the protease a long way from the two bound MepS molecules. Is it possible that the interaction between MepS and NlpI is regulatory rather than a site that serves as a waiting area for being chopped up by the protease?

Minor points / suggestions

- The construct diagram that appears in the supplemental data would be better placed in the main manuscript as it is very hard to follow the paper without this to refer to.
- The discussion paragraph on MepH feels out of scope. It could be removed.
- The text describing the structure is often too technical - the structure description could be simplified for clarity without loss of accuracy. Reference to individual atoms is not justified - especially given the modest resolution.
- line 232 talks about the 'carboxamide' of D39 interacting with the guanidinium group of R46. This should read 'carboxyl' group.
However, this sentence is needlessly complicated. The interaction would be much better conveyed by stating that negatively charged D39 is close to positively charged R46.
- Fig 4 can be improved by stating both what is in the syringe and cell directly on the figure.
- Fig 6 the white labels are difficult to read - please recolour for clarity.
- Figure 6B does not carry any information that assists the discussion - it is just a cartoon and very similar to the actual structure that appears in 5A. 6B should probably be removed or significantly altered to convey mechanism.
- On Table S2, please add the K_d values (1/K_a) with an appropriate unit such as μM.
- Table S1 - Please add R_{pim} and R_{meas} to crystallographic data statistics (Table S1) - R_{sym}/R_{merge} is not useful with high redundancy/multiplicity.
- The paper would be improved by removing unwarranted abbreviations. For example 'EPase' can be replaced with 'endopeptidase'.
- Figure S16 and S19 appear to have density maps that are carved too close to the model. Please check whether these figures might be improved.
- Figure S21 - please label the image directly with the protein names (NlpI, MepS etc).
- The authors are commended for recognising the quality of the density in the main text, and for supplying .PDB and .MTZ files for review.

Reviewer #3:

Remarks to the Author:

In this manuscript, the authors present the structure of two complexes formed between proteins that are involved in the synthesis and elongation of peptidoglycan (PG) in bacterial cells. The structural data are interesting and proposes a model of colocalization and recruitment of these proteins. However, some mutational and in vivo assays, or interaction data with PG sacculi or chains are required to confirm the proposed mechanisms.

Below are listed some other points that should be reviewed or commented.

- Could you comment the stoichiometry of 1 obtained in the ITC experiments, while the X-ray structure shows a hexamer with a dimer of NlpI bound to four mMepS?
- How the authors can be sure that the hexameric structure obtained by X-ray does exist in solution and is not an artefact of the crystal? Is there any evidence of this stoichiometry and binding mode

in solution?

- NIpI is a lipoprotein. There is no information on the preparation of this (lipo)protein. Is NIpI in soluble form or in lipo-form in the binding experiments? Could you provide clarification regarding NIpI preparation in the context of binding experiment. Ref 54 (Line 382) does not mention the preparation of this protein.

-Line 365: How is the sequence conservation for the disordered N-terminus? Is there any pattern?

Line 250: This is not clear. May be a figure can be added to better explain this statement.

Line 286: Elaborate more

-Overall, the figures and their legends need a thorough revision to ensure the clear understanding of the data.

-The purpose of Figure S2 is not clear.

-The spectral superpositions are not always well visible. For example in Figure 1B and 1E.

-In S3, I cannot see the difference between blue and black.

-Figure S6A: Add "holoform" or "NIP-bound state"

Line 130: I see two peaks moving, please add a csp per residue plot and comment on these changes.

Line 134: Linewidth were hardly observed. Please rephrase to make it understandable for non-NMR experts.

Line 137: Do carbon chemical shift indicate any secondary structure propensity?

Figure 1E: Name the peaks that are less effected and explain.

Figure S3: add error bars

Figure 3 and 5: please add the localization of where the lipidation of NIpI would occur

Line 176-190: all this detail can be shortened, Fig 3C,D,E,F can go into the SI

Line 191-208: same, the location of salt bridges can be moved to the SI. Please Simplify.

Line 220-236: again, please simplify these details.

Line 238: The authors mention that "conformational changes occur as peaks become broadened".

How are you sure that the line broadening stems from Rex as in an exchange contribution to R2 and not merely from the complex being very large?

Line 244: Add errors to these ratios based on the noise in the spectra.

REVIEWER COMMENTS

Reviewer #1 (Remarks to the Author):

The structural and biochemical study by Shen Wang and company presents the structure of the NlpI adaptor complexed with the endopeptidase MepS, revealing how NlpI recruits and influences multiple MepS molecules, promoting PG expansion. NlpI binding induces a structural change in the intrinsically disordered N-terminal of MepS, promoting MepS dimerization and enhanced activity in PG hydrolysis. The protein level of MepS binds to NlpI. The structure of the Prc-NlpI-MepS complex demonstrates that NlpI facilitates the interaction between MepS and Prc, leading to efficient degradation of MepS by Prc. The findings shed light on how NlpI enhances the binding efficiency of cellular endopeptidases and directs MepS degradation by Prc. Please see my comments below.

Major Concerns

1. Understanding the significance of the study and the advancements it made in the context of existing knowledge was difficult to grasp on a global scale based solely on this paper.

• Suggestion: The introduction should be expanded to provide more context and clearly state the study's contributions and how they advance the current knowledge in the field.

We thank the reviewer for bringing out this point. As suggested, we have revised the introduction and discussion accordingly.

2. The study focuses on the binding interactions between NlpI, PrcSK, and mMepS mutants but does not explore the broader context of other potential interacting partners mentioned in lines 78-82 and again 329-330.

• Suggestion: Acknowledge the limited scope and suggest avenues for future research to investigate other potential interacting partners and their relevance.

We thank the suggestion and have revised the introduction and discussion accordingly.

3. The study is mainly based on in vitro experiments, lacking in vivo validation of mutants, etc., to understand and demonstrate the relevance of the findings in a more physiological setting.

We appreciate the reviewer comments and suggestions. To examine how the N-terminal of MepS affected the association of NlpI with MepS and/or Prc in vivo, we developed an antibody for NlpI and overexpressed MepS-N53 using an arabinose-inducible plasmid that encodes residues 1-53. We then investigated cell morphological changes, cell envelope integrity, and the cellular mMepS level through immunoblot analysis of a chromosomal MepS-3XFlag derivative. According to previous studies, the level of mMepS is quite plentiful during the log phase, dropping abruptly in the stationary phase. However, overexpression of MepS-N53 causes cell morphological changes, including the formation of long filaments, and leads to the accumulation of MepS throughout the exponential and stationary phases (Fig 6). Furthermore, the prolonged filamentation induced by MepS-N53 adversely affects cell wall stability, as demonstrated by an envelope integrity assay utilizing red- β -D-galactopyranoside (CPRG) (Fig 6C). Our results suggest that the cellular MepS level is substantially disturbed by the N-terminal of MepS, while the protein level of NlpI, detected by NlpI-specific antibody, is not reduced in the overexpressing MepS-N53 strain. These findings provide physiological evidence that the N-terminal of MepS impacts the morphogenesis of *E. coli* by altering the interaction between NlpI and MepS.

4. The extent of the proposed model's applicability to other bacterial systems remains uncertain or unclear from the work presented.

• Suggestion: Discuss the limitations of the proposed model and consider how it might be tested in other bacterial systems to establish its broader applicability.

We thank the reviewer suggestion and have modified discussion accordingly. We examined the conservation of the MepS and NlpI structures, respectively, through the sequence alignments of the similar sequences identified through BLAST search (Supplementary Figs. 22-23), suggesting that the homologs of MepS and NlpI both share a common tertiary and quaternary organization. Based on the Protein Disorder prediction System¹, the N-terminals of the MepS homologs contain intrinsically disordered regions. Analysis of the sequence of MepS N-terminal showed

conservation of the hydrophobic residues L24 and F31, which are involved in the interaction with NlpI (Supplementary Fig. 22), implicating that the *E. coli* MepS-NlpI interaction may occur and form similar complexes in other Gram-negative species. Therefore, the complex structures potentially inform on important aspects of cell envelope biology in Gram-negative bacteria.

Ref:

1. Ishida, T. & Kinoshita, K. PrDOS: prediction of disordered protein regions from amino acid sequence. *Nucleic Acids Res* **35**, W460-464.

5. The authors should have included controls where the individual proteins (NlpI, Prc^{SK}, and mMepS) are tested independently to confirm their binding and interaction, and also test the mutants in isolation to verify their effects on binding.

The point is well taken. Upon careful examination of all the control experiments, it became evident that the ITC results displayed almost no heat absorption or release for the control sample. The ITC and SEC experiments were performed and are detailed below:

SEC experiments

1. NlpI alone (Figure 2C)
2. mMepS alone (Figure 2C)
3. F31A alone (Figure 2C)
4. dN36 alone (Figure 2C)
5. NlpI+ mMepS (Figure 2C)
6. NlpI+ F31A (Figure 2C)
7. NlpI+ dN36 (Figure 2C)

ITC experiments

1. NlpI alone (Supplementary Fig. 25)
2. mMepS alone (Supplementary Fig. 25)
3. MepS-F31A alone (Supplementary Fig. 25)
4. MepS-L24R alone (Supplementary Fig. 25)
5. MepS-D39A alone (Supplementary Fig. 25)

6. MepS-Q28A alone (Supplementary Fig. 25)
7. dN36 alone (Supplementary Fig. 25)

6. The crystal structures (3.5 Å for PrcSK-NlpI-mMepS and 2.8 Å for NlpI:mMepS complex) do not fully support all the conclusions drawn in the study (see below)

We sincerely appreciate the reviewer suggestion and have promptly updated Figure 5A and Supplementary Fig.16 accordingly (see below). We would be delighted to provide the PDB and MTZ files for your thorough review.

7. The electron density of the PrcSK-NlpI-mMepS complex in Figure 5A is not convincing and does not clearly show the complex at the defined resolution. The maps are barely interpretable.

Thank you for pointing this out to us. Figure 5A is updated in the revised manuscript.

8. The electron density map for Figure S16 is also not convincing.

We appreciate the reviewer comments. In the revised version of the manuscript, Supplementary Fig. S16 has been updated.

9. The quality of chains A through L in the PDB validation for the Crystal structure of adaptor NlpI in complex with endopeptidase MepS and PDZ-protease Prc is poor for the reported resolution.

Firstly, I would like to express my sincere gratitude to reviewer 1 for raising the issue regarding chain quality. The detailed response is provided below.

Following the major revision, we attempted to optimize the original crystallization conditions and collect new diffraction data. However, the quality and resolution of the subsequently obtained diffraction data did not surpass that of the original data with a resolution of 3.5 Å. The failure to enhance the resolution may be attributed to the much higher Matthews coefficient and solvent content of the crystal (approximately 4.2 and 0.71) than usually seen, indicating very

loose crystal packing. Consequently, it is difficult to obtain higher resolution data from this form of complex crystal.

In fact, not all chains exhibit poor quality. Based on the structure validation report by wwPDB, only Chain C (Prc) and D (Prc) have lower quality compared to the other chains (A, B and I to L). This may be attributed to the crystal's symmetry (P43212) and packing, which result in fewer contacts and lead to numerous missing electron densities for Prc. However, in the regions near NlpI and MepS, the electron density map for Prc is actually quite good. On the other hand, by comparing the structure validation results of the other chains with those of the NlpI-MepS crystal at a resolution of 2.8 Å, it is clear that the quality of the other chains is also good.

The reason why this data set cannot achieve a high resolution and has relatively poor chain quality for Prc may be attributed, in part, to the fact that the complex structure is primarily composed of multiple chains of different proteins, including two NlpI (297 residues), two Prc (688 residues) and four MepS (168 residues). In order to make a comparison, we searched the Protein Data Bank for similar complex structures (composed of multiple molecules with resolution falling around 3.5 Å) to appraise their chain quality. The results revealed that many complex structures deposited in 2023 share similar characteristics of chain quality with ours. Examples include 7DRR (3.48 Å; 4 proteins), 8R2G (3.45 Å; 15 proteins), 8HNV (3.41 Å; 4 proteins), 8H2U (3.4 Å; 19 proteins), 8F2V (3.5 Å; 6 proteins), 8BUP (3.41 Å; 9 proteins) and 8ACW (3.4 Å; 6 proteins), among others. Therefore, while our data set exhibits relatively poor quality in two chains compared to other deposited complex structures, it still falls within an acceptable range.

Minor Concerns

1. *Lines 116-117: "The intrinsically disordered N-terminal of endopeptidase MepS is crucial for the interaction with adaptor NlpI." appears to be an overstatement.*

We thank the reviewer suggestion and have updated the sentence as follows:

The intrinsically disordered N-terminal region of endopeptidase MepS is involved in the interaction with adaptor NlpI.

2. In lines 119-120: "The NlpI-interacting EPase MepS has been reported to adopt a papain-like $\alpha+\beta$ fold with a novel catalytic triad comprising C68, H119, and H13142,44 . This sentence lacks clarity and precision. The information provided is not clear in explaining how the "papain-like $\alpha+\beta$ fold" and the "novel catalytic triad" relate to the interaction with NlpI.

In the revised version of the manuscript, we have improved the sentence as follows:

It has been reported that MepS adopts a papain-like $\alpha+\beta$ fold, featuring with a catalytic triad comprising C68, H119 and H131.

3. Lines 123-127: "Based on the reported NMR structure of MepS⁴², truncated mutant dN36-MepS (residues 37-162) was selected for our initial titration experiments (Figure S1), while the first 36 N-terminal residues, which were previously suggested to be significantly disordered in the screening by NMR⁴², was suggested to be dispensable for a catalytic mechanism in the cysteine peptidase family." The sentence structure is awkward and may lead to confusion. It is not clear what the authors are trying to convey regarding the relevance of the N-terminal residues in the catalytic mechanism.

We thank the reviewer for this recommendation. In the revised version of the manuscript, we have improved the sentence as follows:

Based on the reported NMR structure of MepS, we chose to initiate the titration experiments on the truncated mutant dN36-MepS (as illustrated in **Fig. 1A**), which includes residues 37 to 162. Notably, the N-terminal segment exhibited significant disorder and was thus excluded from both NMR and X-ray structure determinations published earlier^{2,3}.

Ref:

2. Aramini, J. M. *et al.* Solution NMR structure of the NlpC/P60 domain of lipoprotein Spr from *Escherichia coli*: structural evidence for a novel cysteine peptidase catalytic triad. *Biochemistry* **47**, 9715-9717.

3. Lee, W. C., Jang, A., Lee, J. Y. & Kim, Y. Structural implication of substrate binding by peptidoglycan remodeling enzyme MepS. *Biochem Biophys Res Commun* **583**, 178-183.

4. Lines 131-133: "Interestingly, the NMR spectra of dN36-MepS in the absence and presence of unlabeled NlpI at a molar ratio of 1:1 (Fig. 1B) were very similar, suggesting that NlpI scarcely binds to dN36-MepS." The authors state that NlpI scarcely binds to dN36-MepS based on the similarity of NMR spectra in the absence and presence of NlpI. However, this conclusion lacks a clear quantitative analysis or statistical evidence to support it in the text.

The point is well taken. We have included quantitative analyses (see Supplementary Figs. 2 and S3), and the results are explained in more detail as follows:

The attenuation of the peak intensity was measured using two-dimensional ^{15}N - ^1H NMR spectroscopy, and the ratios of signal intensity for the dN36-MepS and mMepS were reduced by approximately $23 \pm 7\%$ and $93 \pm 5\%$, respectively (Supplementary Figs. 2 and 3). This suggests that the N-terminal disordered region of mMepS greatly contributes the binding to NlpI.

5. Lines 135-143: "We next found that NlpI prominently induced dramatic variations in the 2D spectral features of mMepS and the linewidth of the bound-state resonances was hardly observed upon NlpI binding (Fig. 1C)." This statement was unclear and needs additional details or clarification regarding the nature of the "dramatic variations" and "linewidth of the bound-state resonances."

We thank the reviewer for this suggestion. In the revised version of the manuscript, we have improved the sentences as follows:

Interestingly, the NMR spectra of dN36-MepS in the absence and presence of unlabeled NlpI at a molar ratio of 1:1 (**Fig. 1B**) were very similar, while the peak signals of mMepS were almost disappeared and showed an overall line-broadening upon addition of NlpI (**Fig. 1C**). The attenuation of the peak intensity was measured using two-dimensional ^{15}N - ^1H NMR spectroscopy, and the ratios of signal intensity for the dN36-MepS and mMepS were reduced by approximately $23 \pm 7\%$ and $93 \pm 5\%$, respectively (Supplementary Figs. 2 and 3). This suggests that the N-terminal disordered region of mMepS greatly contributes the binding to NlpI.

Afterwards, we tested the importance of N-terminal disordered region by preparing the truncated mutants MepS-N39 (residues 1-39) and MepS-N53 (residues 1-53) (**Fig. 1A**). The spectra of MepS-N39 and MepS-N53 exhibited highly disordered peptide segments (**Fig. 1E**), indicated by rather intense peaks with poor dispersion around 7.5-8.5 ppm of the proton dimension. Titration of unlabeled NlpI into ¹⁵N-labeled MepS-N39 and MepS-N53, respectively, caused substantial peak broadening (**Fig. 1F**). The results demonstrated that, despite the lack of the core structure, the intrinsically disordered N-terminal of MepS still retains the binding ability for adaptor protein NlpI, confirming that it has a key role in the interaction with NlpI.

7. Lines 213-219: "Compared with WT mMepS, the truncated dN36-MepS mutants showed no heat changes with the titration of NlpI, suggesting that the first 36 N-terminal residues prominently contribute to NlpI binding (Fig. 4B and Table S2)." The statement suggests that the first 36 N-terminal residues "prominently" contribute to NlpI binding. The phrase "no heat changes" needs further explanation and clarification.

We appreciate the comments and suggestions. The sentence has updated as follows:
Compared with WT mMepS, the NlpI titration into truncated dN36-MepS mutants does not cause heat absorption or release, suggesting that the N-terminal residues prominently contribute to an enthalpically favorable reaction toward NlpI binding.

8. Lines 260-262: "To understand how NlpI enhances the efficiency of mMepS proteolysis by Prc, we determined the crystal structure of PrcSK-mMepS-NlpI at 3.5 Å, revealing that two monomeric PrcSK bind to hetero-hexameric NlpI-mMepS to form a 2:2:4 hetero-octameric complex (Fig. 5A)." The authors state that the complex formed is a "2:2:4 hetero-octameric complex" may need more evidence or justification for this particular stoichiometry. More details and analysis might be needed.

We also examined the stoichiometry of the Prc^{SK}-mMepS-NlpI complex through simultaneous SAXS and UV-Vis absorption measurements. As demonstrated in previous studies⁴, SEC-SAXS could determine the composition of a protein complex through combined analysis of the zero-angle scattering intensity of SAXS and UV-absorption intensity. The results showed that two

monomeric Prc^{SK} molecules bind to hetero-hexameric NlpI-mMepS, forming a 2:2:4 hetero-octameric complex measured at the eluent peak, with a confidence level exceeding 90% (**Fig. 5B**). The detailed quantitative analysis was shown in the **Supplementary Note 1**.

Ref:

3. Shih, O. *et al.* Membrane Charging and Swelling upon Calcium Adsorption as Revealed by Phospholipid Nanodiscs. *J Phys Chem Lett* **9**, 4287-4293.

Reviewer #2 (Remarks to the Author):

Wang and colleagues present two fabulous new crystal structures of the outer membrane lipoprotein NlpI bound to MepS and protease Prc. The structures are interesting and potentially inform on important aspects of cell envelope biology in *E. coli*.

The co-structure of the NlpI/MepS/Prc is particularly impressive and will certainly help the field to understand the interactions of NlpI. The complex also defines a structural basis on which to build further understanding of MepS endopeptidase degradation in the future.

There is some very nice supporting data that documents the importance of the MepS N-terminus in the interaction with NlpI. The NMR experiments support a disorder-to-order transition in the MepS N-terminus during NlpI binding. The use of multiple techniques to further investigate the NlpI/MepS interaction is commendable.

However, there are some concerns over the supporting biochemical data (and interpretation) that should be addressed. In particular, the SEC, SPR and ITC show inconsistencies that need to be resolved. Fortunately, these data most likely can be clarified by appropriate repeats and inclusion of missing controls.

Some comments below.

1. For the size exclusion data, the authors perform an NlpI-binding experiment in competition between full length MepS (mMepS) and MepS lacking its N-terminus (dN36MepS). While the competition experiment is welcome, the various simpler controls should also be performed and reported. For example, there should be runs of each individual protein (mMepS alone, dN36MepS alone, NlpI alone) as controls, and non-competitive pairwise binding experiments - ie mMepS+NlpI, and dN36MepS+NlpI. It will be particularly interesting to see whether MepS

lacking its N-terminus (dN36MepS) can still bind NlpI (absent of competition with the wild type). These experiments should also help resolve the disparity between the SPR and ITC results (see below).

We deeply appreciate the reviewer comments and suggestions. Upon careful examination of all the control experiments, it became evident that the ITC results displayed almost no heat absorption or release for the control samples. The ITC and SEC data have been added to the manuscript, including:

SEC experiments

NlpI alone (Figure 2C)

mMepS alone (Figure 2C)

F31A alone (Figure 2C)

dN36 alone (Figure 2C)

NlpI+ mMepS (Figure 2C)

NlpI+ F31A (Figure 2C)

NlpI+ dN36 (Figure 2C)

ITC experiments

NlpI alone (Supplementary Fig. 25)

mMepS alone (Supplementary Fig. 25)

MepS-F31A alone (Supplementary Fig. 25)

MepS-L24R alone (Supplementary Fig. 25)

MepS-D39A alone (Supplementary Fig. 25)

MepS-Q28A alone (Supplementary Fig. 25)

dN36 alone (Supplementary Fig. 25)

2. The ITC data follows up on the structural observations very well. The authors identify a number of interface residues from the structure(s) and then measure pairwise interactions between MepS and NlpI for each mutation. The data shows that the MepS N-terminus is especially important for binding. The data also suggests that the F31A mutation completely

breaks the interaction - which is consistent with F31 being located in the MepS N-terminus.
However, no repeats are reported - so these are presumably single ITC measurements, and the
main text overanalyses the very small differences between Kd values measured. (Using Table S2,
the measured Kd values are: 0.2 uM, 1 uM, 3 uM, 6 uM, 2 uM, and 0.3 uM.)
The Kd values in the text are reported with tremendous precision (ie two decimal places on the
uM measurements - ie 10 nM precision). The results are then discussed as if this level of
accuracy were real.
Clearly, the N-terminal deletion and F31A mutant affect binding - but the other Kd values are
probably not far outside experimental error. L24R, Q28A and D39A are all very similar
measurements - and there is no justification to draw any insight from closely comparing these
values.
The paper would be better served if it discussed the Kd measurements as minimal changes to
binding affinity for all mutants except F31A and the N-term deletion. It should also report the
measured Kd values in Table S2 (not just Ka) and only give values to the nearest uM in the text.
There should also be at least one further repeat for mMepS (the WT), dN36MepS, and MepS-
F31A, before these results are considered reproducible. In fact, if the WT were to be measured
three times, it would give a much more realistic impression of experimental variability for this
method and better inform the meaning of the small differences in Kd observed between WT and
most mutants.

We greatly thank the reviewer for bringing out these points. As suggested, we have updated Supplementary Table S2, which now includes triplicate data for mMepS (the WT), dN36-MepS, N53, N39, MepS-L24R, MepS-Q28A, MepS-F31A and MepS-D39A. Because the heat changes in the binding events are monitored in ITC experiment, and only the values of ΔH and K_D with standard deviations are obtained by fitting ITC profiles. Both ΔG and ΔS are indirectly determined by two equations: (1) $\Delta G = RT \ln K_D = -RT \ln K_A$; (2) $\Delta G = \Delta H - T\Delta S$. The averages and standard deviations of each thermodynamic parameters for three independent experiments were listed in Supplementary Table S2. All the Kd values mentioned in the manuscript have been updated with averages of triplicate data with standard deviations. We also have updated the results in the text as follows:

To validate the above structural observations, we measured the affinity of NlpI to the mMepS mutants by isothermal titration calorimetry (ITC) experiments. In the case of NlpI binding to wild-type (WT) mMepS (**Fig. 4A and Supplementary Table S2**), we obtained the enthalpy-driven interaction with the K_D $0.24 \pm 0.02 \mu\text{M}$, which was consistent with previously published results. Compared with WT mMepS, the NlpI titration into the truncated dN36-MepS mutants does not cause an enthalpically favorable binding reaction, suggesting that the first 36 N-terminal residues prominently contribute to NlpI binding (**Fig. 4B and Supplementary Table S2**). By contrast, we investigated the affinity of the truncated mutants MepS-N53 and MepS-N39 for NlpI; the results revealed that both interactions were enthalpically favorable with the K_D of 1.45 ± 0.22 and $2.68 \pm 0.15 \mu\text{M}$, respectively, but with the 6-11 fold differences in binding affinity (**Fig. 4C-D and Supplementary Table S2**). We also chose to mutate key residues at the NlpI–mMepS interface inferred from our X-ray ternary structure. For example, the hydrophobic side-chain of L24 of mMepS-1 has multiple contacts with the side-chains of R78', A79', R82', Q108' and A109', and the aromatic ring of Y105' located at the h3'-h4' of NlpI'; mutation L24R binds to NlpI dimer with an approximate 28-fold increase in the K_D value (**Fig. 4E and Supplementary Table S2**). The side-chain of mMepS-1 Q28 has contacts with the side-chains of residues R82', N83', and S86' located at the h3' of NlpI'; mutation Q28A resulted in a decreased binding affinity by a factor of 7.7 (**Supplementary Fig. 11 and Supplementary Table S2**). The aromatic ring of mMepS-1 F31 has hydrophobic contacts with NlpI h1 (involved with the side-chains of L38, V42 and A45) and NlpI' h3' (involved with the side-chains of A79' and N83') located at the dimerization interface of NlpI homodimer. Interestingly, the NlpI titration to mutation F31A had no sufficient heat release or absorption to allow the K_D determination (**Fig. 4F and Supplementary Table S2**), indicating that the bulky hydrophobic side-chain of F31 plays a dominant role in the NlpI–mMepS interaction. The side-chain of mMepS-1 D39 has polar contacts with residue R46 of NlpI h1; the K_D of D39A variant for NlpI was almost unchanged ($0.39 \pm 0.11 \mu\text{M}$), and there seems to be enthalpy-entropy compensation that leads to very little or no effect in the free energy changes (**Supplementary Fig. 11 and Supplementary Table S2**). In summary, we conducted ITC experiments to examine the binding of NlpI with MepS mutants, including dN36-MepS, N53, N39, MepS-L24R, MepS-Q28A, MepS-F31A and MepS-D39A. The enthalpic contribution is notably influenced by the N-terminal of MepS, where the residues (Q28 and D39) involved in the hydrophilic interactions

result in little or no difference in the binding affinity, while the hydrophobic residues (L24 and F31) are the main determinants for association with NlpI.

3. The Surface Plasmon Resonance (SPR) data is not consistent with ITC results - even though it is reported as if the two are in step.

The SPR gives a much higher affinity than ITC for NlpI and the WT MepS (mMepS) at 0.03 μ M, but also suggests that the MepS lacking the N-terminus, or incorporating the interface-breaking F31A mutant, still have K_d values \sim 4.75 μ M.

A K_d value of 5 μ M is similar to (or even slightly tighter than) the affinity between EDTA and calcium. Thus SPR suggests the mutants are still forming a tight complex with NlpI.

The K_d values reported by SPR are therefore very different to the ITC. The ITC data implies F31A and MepS lacking the N-terminus are essentially non-binding - while the SPR suggests quite a tight interaction ($K_d < 5 \mu$ M). The ITC should be repeated to check that the non-binding results are correct, and further controls should be performed for the SPR to ensure these are not measuring something other than the protein:protein interaction.

Reporting the SEC for NlpI/MepS, NlpI/dN36MepS and NlpI/MepS-F31A would also be extremely useful to check whether the ITC or SPR is more likely to be correct.

We sincerely appreciate your valuable comments and suggestions. To understand the discrepancy between the SPR data and ITC results, we conducted additional experiments as suggested. Upon careful examination of all the results, it became evident that the ITC results displayed no heat absorption or release for the control samples. However, in the SPR data, we observed non-specific interactions of mMeps with the CM5 chip, which contributed to the misinterpretation of the results. Therefore, we decided to remove all the SPR data from our manuscript. Furthermore, we performed the SEC experiments to investigate the interactions of NlpI with mMepS mutants. The SEC chromatograms of both F31A and dN36 mutants confirmed that disrupting the interaction between the N-terminal of MepS and NlpI would compromise the NlpI-mMepS assembly as the mutants F31A and dN36 were both eluted as separate peaks in the present of NlpI (**Fig. 2C**). On the contrary, the NlpI:mMepS complex was stably formed as demonstrated by the shifted protein peak with a larger apparent size (**Fig. 2C**).

4. Finally for the discussion, the architecture of the NlpI/MepS/Prc complex seems to orient the protease a long way from the two bound MepS molecules. Is it possible that the interaction between MepS and NlpI is regulatory rather than a site that serves as a waiting area for being chopped up by the protease?

We thank the suggestion and have revised the discussion accordingly. During the exponential growth phase, the role of NlpI is to recruit the endopeptidase MepS and bring about the avidity effect toward PG binding. This suggests that the interaction between MepS and NlpI might have a regulatory function, and the binding may have a controlling or modulating role in the cellular processes involving these proteins, rather than serving as a mere waiting area for protease-mediated degradation.

Unfortunately, the ligand-binding PDZ domain of Prc^{SK} is hardly defined in the density map of the Prc^{SK}-NlpI-MepS complex (Supplementary Fig. 18B), suggesting that the PDZ domain is highly dynamic. Owing to the missing electron density of PDZ domain, neither the position of PDZ domain or the distance between the C terminus of mMepS and PDZ domain can be clearly determined. However, our complex structure still reveals the C terminus of MepS2 faces to the concave surface formed by the h2-h6 of NlpI (Supplementary Fig. 20), while that of mMepS-1 is solvent-exposed (Supplementary Fig. 19), suggesting that Prc might first target mMepS-1 for degradation rather than mMepS-2 according to the localization of their C-terminal tails. Furthermore, the binding of MepS with NlpI becomes weaker in the presence of Prc. Consequently, MepS is able to move or rotate within the cradle formed by NlpI and Prc at the stationary phase, providing the opportunity for the flexible C-terminus of MepS to reorient toward the PDZ domain of Prc.

Minor points / suggestions

- The construct diagram that appears in the supplemental data would be better placed in the main manuscript as it is very hard to follow the paper without this to refer to.

We thank the suggestion and have revised the figure accordingly. The construct diagram has been shown in Fig 1A.

- The discussion paragraph on MepH feels out of scope. It could be removed.

We thank you for this point. The discussion paragraph on MepH is removed.

- The text describing the structure is often too technical - the structure description could be simplified for clarity without loss of accuracy. Reference to individual atoms is not justified - especially given the modest resolution.

We thank the suggestion and have revised the text accordingly.

- line 232 talks about the 'carboxamide' of D39 interacting with the guanidinium group of R46. This should read 'carboxyl' group.

However, this sentence is needlessly complicated. The interaction would be much better conveyed by stating that negatively charged D39 is close to positively charged R46.

The point is well taken and have revised the text accordingly.

- Fig 4 can be improved by stating both what is in the syringe and cell directly on the figure.

We thank you for this point and have revised the figure accordingly.

- Fig 6 the white labels are difficult to read - please recolour for clarity.

We thank you for this point and have revised the figure accordingly.

- Figure 6B does not carry any information that assists the discussion - it is just a cartoon and very similar to the actual structure that appears in 5A. 6B should probably should be removed or

significantly altered to convey mechanism.

We thank the suggestion and have revised the figure and discussion accordingly.

- On Table S2, please add the Kd values (1/Ka) with an appropriate unit such as uM.

We thank you for this point and have revised Supplementary Table S2 accordingly.

- Table S1 - Please add Rpim and Rmeas to crystallographic data statistics (Table S1) - Rsym/Rmerge is not useful with high redundancy/multiplicity.

We thank you for this point and have revised the crystallographic data statistics accordingly.

- The paper would be improved by removing unwarranted abbreviations. For example, 'EPase' can be replaced with 'endopeptidase'.

We thank you for this point and have revised the figure accordingly.

- Figure S16 and S19 appear to have density maps that are carved too close to the model. Please check whether these figures might be improved.

We thank you for this point and have revised the figures accordingly. Supplementary Fig. 19 is moved to S18.

- Figure S21 - please label the image directly with the protein names (NlpI, MepS etc).

We thank you for this point and have revised the figure accordingly. Supplementary Fig. 21 is moved to Supplementary Fig. 24.

- The authors are commended for recognising the quality of the density in the main text, and for

supplying .PDB and .MTZ files for review.

Reviewer #3 (Remarks to the Author):

In this manuscript, the authors present the structure of two complexes formed between proteins that are involved in the synthesis and elongation of peptidoglycan (PG) in bacterial cells. The structural data are interesting and proposes a model of colocalization and recruitment of these proteins. However, some mutational and in vivo assays, or interaction data with PG sacculi or chains are required to confirm the proposed mechanisms.

Below are listed some other points that should be reviewed or commented.

1. Could you comment the stoichiometry of 1 obtained in the ITC experiments, while the X-ray structure shows a hexamer with a dimer of NlpI bound to four mMepS? How the authors can be sure that the hexameric structure obtained by X-ray does exist in solution and is not an artefact of the crystal? Is there any evidence of this stoichiometry and binding mode in solution?

The point is well taken. We also performed size exclusion chromatography coupled with small angle x-ray scattering (SEC-SAXS) experiments to examine the stoichiometry of the complex structure of NlpI-mMepS. The analysis revealed that the SAXS data acquired for NlpI-mMepS complex resulted in an R_g of 35.1 Å. Moreover, the SAXS profile of NlpI-mMepS is consistent with the crystallographic results, with $\chi^2 = 1.73$, confirming that the hexameric structure does exist in solution (**Fig 3B**).

In accordance of the NlpI-mMepS complex structure, the N-terminal region of one protomer (mMepS-1) prominently engages with the dimerization interface of NlpI while that of the other protomer (mMepS-2) only interacts with one of NlpI. This suggests that the dimerization interface of NlpI is crucial for mMepS-1 but not mMepS-2 (**Supplementary Fig. 8**). To explain the the stoichiometry of 1 obtained in the ITC experiments, we used a monomeric mutant NlpI- ΔN (T37-Q294) to characterize the interactions by NMR and ITC experiments. Titration of unlabeled NlpI- ΔN into ^{15}N -labeled mMepS caused line-broadening from residues 26-162 but not include residues 1-25 (**Supplementary Fig. 13A**). The result is in agreement with the crystal structure, as residues 23-25 interacts with NlpI. Furthermore, the interaction between mMepS and NlpI- ΔN does not exhibit the exothermic or endothermic signals monitored via ITC

(**Supplementary Fig. 13B**), while the binding of NlpI- Δ N to mMepS was directly observed in the NMR experiments. Therefore, we conclude that only mMepS-1 interacts with NlpI dimer with an enthalpically favorable binding response as mMepS-2 could bind to NlpI without apparent heat release or absorption, thereby a lower stoichiometry was observed in the ITC binding reaction.

2. NlpI is a lipoprotein. There is no information on the preparation of this (lipo)protein. Is NlpI in soluble form or in lipo-form in the binding experiments? Could you provide clarification regarding NlpI preparation in the context of binding experiment. Ref 54 (Line 382) does not mention the preparation of this protein.

Thank you for catching this. Previous reports have experimentally characterized the lipidation of NlpI, which occurs at residue C19. Our construct does not include residue C19. The NlpI sequence (residues S20-Q294) without the signal peptide was cloned into a protein-expressing vector. Purified mature NlpI is soluble to at least 200 mg/ml in 10 mM Tris/HCl pH 8.0, 10 mM NaCl. The subsequently purified NlpI was used for protein crystallization, preventing us from observing the localization of lipidation in our protein complex structure. Therefore, we have included a schematic diagram of the lipid-anchor site of NlpI in our model to illustrate the characteristics of the lipoprotein. Furthermore, author list of references 35 and 49 have been updated in the revised manuscript.

3. Line 365: How is the sequence conservation for the disordered N-terminus? Is there any pattern?

We examined the conservation of the MepS and NlpI structures, respectively, through the sequence alignments of the similar sequences identified through BLAST search (**Supplementary Figs. 22-23**), suggesting that the homologs of MepS and NlpI both share a common tertiary and quaternary organization. Based on the Protein DisOrder prediction system¹, the N-terminals of the MepS homologs contain intrinsically disordered regions. Analysis of the sequence of MepS N-terminal showed conservation of the hydrophobic residues L24 and F31, which are involved in the interaction with NlpI (**Supplementary Fig. 22**), implicating that the *E. coli* MepS-NlpI

interaction may occur and form similar complexes in other Gram-negative species. Therefore, the complex structures potentially inform on important aspects of cell envelope biology in Gram-negative bacteria.

4. Line 250: This is not clear. May be a figure can be added to better explain this statement.

We appreciate the suggestion and have updated **the text** as follows:

The escalating concentrations (25-100 μM) of unlabeled NlpI dimer resulted in the attenuation of NMR signals from the ^{15}N -labeled F31A variant (50 μM). Notably, the impact was more pronounced on the core structure than the intrinsically disordered N-terminal region (Fig. 4G), highlighting the essential role of the hydrophobic aromatic ring of F31 in the interaction with NlpI.

5. Line 286: Elaborate more

We thank the suggestion and have updated **the text** as follows:

Moreover, in each PrcSK, we observe a co-purified peptide in the proteolytic site, with the unidentified peptide represented by a poly-Ala model (**Supplementary Fig. 19**). The locations and orientations of peptide fragments show a high similarity to those in the structure of the NlpI-Prc-K477A complex.

6. Overall, the figures and their legends need a thorough revision to ensure the clear understanding of the data.

We thank the suggestion and have revised the legends accordingly.

7. The purpose of Figure S2 is not clear.

We thank the suggestion and have updated **the text** accordingly.

8. The spectral superpositions are not always well visible. For example in Figure 1B and 1E.

We thank the suggestion and have updated **the figures** accordingly.

9. In S3, I cannot see the difference between blue and black.

We thank the suggestion and have updated **the figures** accordingly and moved to Supplementary Fig. 4.

10. Figure S6A: Add “holoform” or “NIP-bound state”

We thank the suggestion and have updated **the figures** accordingly.

11. Line 130: I see two peaks moving, please add a csp per residue plot and comment on these changes.

The NMR spectra of dN36-MepS (residues 37-162) and mMepS (residues 1-162) exhibit a high degree of superimposition, except for residues 37-45, 69, 81, 105, and 160 (Supplementary Fig. **1A**). A chemical shift perturbation (CSP) per residue plot is incorporated into the revised manuscript (**Supplementary Fig. 1**), while approximately 90% of the backbone resonances of mMepS were assigned through the utilization of multidimensional heteronuclear NMR experiments (**Supplementary Fig. 1B**).

12. Line 134: Linewidth were hardly observed. Please rephrase to make it understandable for non-NMR experts.

We thank the suggestion and have updated **the text** as follows:

The NMR spectra of dN36-MepS in the absence and presence of unlabeled NlpI were very similar (**Fig. 1B**), while the peak signals of mMepS were almost disappeared and showed an overall line-broadening upon addition of NlpI (**Fig. 1C**). The attenuation of the peak intensity was measured using two-dimensional ^{15}N - ^1H NMR spectroscopy, and the ratios of signal intensity for the dN36-MepS and mMepS were reduced by approximately $23 \pm 7\%$ and $97 \pm 7\%$, respectively (**Supplementary Figs. 2-3**). This suggests that the N-terminal disordered region of mMepS greatly contributes the binding to NlpI.

13. Line 137: Do carbon chemical shift indicate any secondary structure propensity?

We thank the reviewer for bringing out this question. Using NMR resonances of the backbone atoms, the $\delta 2D$ algorithm⁵ was employed to determine the secondary structure populations of mMepS. The analysis revealed diminished values in the secondary structure propensities for residues 6-38, with ~3% α -helix, ~5% β -strand, and ~25% PPII (**Supplementary Fig. 1C**), confirming the lack of structural ordering in the N-terminal region of mMepS.

Ref:

5. Camilloni, C., De Simone, A., Vranken, W. F. & Vendruscolo, M. Determination of secondary structure populations in disordered states of proteins using nuclear magnetic resonance chemical shifts. *Biochemistry* **51**, 2224-2231.

14. Figure 1E: Name the peaks that are less effected and explain.

The samples of truncated mutants MepS-N39 (residues 1-39) and MepS-N53 (residues 1-53) are prepared without the treatment of TEV protease and the peaks are from the His-tag and TEV site.

15. Figure S4: add error bars

We thank the suggestion and have updated **the figures** accordingly.

16. Figure 3 and 5: please add the localization of where the lipidation of NlpI would occur

We thank the suggestion and have updated **the figures** accordingly. In *Escherichia coli*, there are several chromosomally encoded lipoproteins with conserved lipobox sequences in their signal peptides. Previous reports have experimentally characterized the lipidation of NlpI, which occurs at residue C19. However, our construct does not include residue C19. The NlpI sequence (residues S20-Q294) without the signal peptide was cloned into a protein-expressing vector. The subsequently purified NlpI was used for protein crystallization, preventing us from observing the

localization of lipidation in our protein complex structure. Therefore, we have included a schematic diagram of the lipid-anchor site of NlpI in our model to illustrate the characteristics of the lipoprotein (**Fig. 7**).

17. Line 176-190: all this detail can be shortened, Fig 3C, D, E, F can go into the SI

We thank the suggestion and have updated **the text** and **the figure** accordingly. Figure_3C-F are moved from the main to the Supplementary Fig. 8.

18. Line 191-208: same, the location of salt bridges can be moved to the SI. Please Simplify.

We thank the suggestion and have updated **the text** and **the figure** accordingly.

19. Line 220-236: again, please simplify these details.

We thank the suggestion and have updated **the text** accordingly.

20. Line 238: The authors mention that “conformational changes occur as peaks become broadened”. How are you sure that the line broadening stems from Rex as in an exchange contribution to R2 and not merely from the complex being very large?

We thank you for this point and the sentence has been removed from the revised manuscript.

21. Line 244: Add errors to these ratios based on the noise in the spectra.

We thank the suggestion and have updated **the figures** accordingly.

Reviewers' Comments:

Reviewer #1:

Remarks to the Author:

The authors have effectively addressed a majority of my concerns, and I find their revisions to be satisfactory. Additionally, they have substantially improved the manuscript. I do not have any further suggestions at this time.

Reviewer #2:

Remarks to the Author:

I am very pleased to see changes to the manuscript addressing the points raised by review.

The revised manuscript addresses experimental concerns on the disparity between SPR and ITC, and reports additional controls and experiments for the Size exclusion/gel filtration that raise confidence in the paper's conclusions.

For the structure of the largest complex, it remains clear that parts of the electron density map are poorly defined, with a low overall resolution. However, this is stated within the manuscript text.

Reviewer #3:

Remarks to the Author:

While the experimental parts of this study represent significant work and involve extensive sample preparations, they lack (even after revision) accurate analysis and presentations of the results. As a result, the conclusions are not clearly explained and not always convincing.

Some major points (among others) are cited below:

In the first version, error bars were absent for the intensity variation of the NMR signal. Although error bars have been added in the current version, the standard deviations are not considered. These are fundamental aspects of NMR spectral analysis.

For other experiments, such as SPR and ITC, basic control experiments were not conducted in the previous versions. Critical information, such as whether the protein was lipidated and which specific domain or mutant was used, is still missing. Instead, the text provides some irrelevant details.

Line 169: Supplementary Figure 4. The spectra superimposition cannot conclusively indicate the same oligomeric state. To determine the oligomeric state from NMR data, the average T1/T2 relaxation time ratio or, at the very least, the peak linewidth should be measured. The statement "the peak linewidth was pretty similar" lacks precision. If the linewidths have been measured, the values should be indicated in the figure.

Line 275: A variation of $0.09 + 0.17$ is not significant. The error is very high. The authors should consider the variation of intensity above the standard deviation. This is valuable for all intensity variation measurements. The whole paragraph is not very clear.

I am not convinced by the explanation on the stoichiometry in the ITC experiments. The authors explain that the NIP1 dimerization interface is crucial for mMepS binding. However, they use a mutant of NIP1 unable to dimerize. The use of this mutant is not mentioned in figures and their legends, nor in the method section, even in the revised manuscript.

The order of supplementary figures is not aligning with their references in the main text, causing difficulty in reading.

In general, the figures and the legends are not clear, the color codes and abbreviations are missing in the legend. A careful analysis of figures and their legends is required.

REVIEWER COMMENTS

Reviewer #1 (Remarks to the Author):

The authors have effectively addressed a majority of my concerns, and I find their revisions to be satisfactory. Additionally, they have substantially improved the manuscript. I do not have any further suggestions at this time.

We sincerely thank the reviewer for the insightful comments, which have greatly improved our revised manuscript. Your feedback has provided a fresh perspective and significantly enhanced the quality of our work. We are glad to hear that the reviewer found our work satisfactory.

Reviewer #2 (Remarks to the Author):

I am very pleased to see changes to the manuscript addressing the points raised by review.

The revised manuscript addresses experimental concerns on the disparity between SPR and ITC, and reports additional controls and experiments for the Size exclusion/gel filtration that raise confidence in the paper's conclusions.

For the structure of the largest complex, it remains clear that parts of the electron density map are poorly defined, with a low overall resolution. However, this is stated within the manuscript text.

We express our sincere gratitude to the reviewer for their critical evaluation and insightful comments, which have significantly enhanced the revised manuscript. We are pleased to know that the reviewer appreciates the effort we put into our paper. Additionally, we would like to convey our appreciation for your understanding regarding the imperfections in our data.

Reviewer #3 (Remarks to the Author):

While the experimental parts of this study represent significant work and involve extensive sample preparations, they lack (even after revision) accurate analysis and presentations of the results. As a result, the conclusions are not clearly explained and not always convincing.

Some major points (among others) are cited below:

1. In the first version, error bars were absent for the intensity variation of the NMR signal. Although error bars have been added in the current version, the standard deviations are not considered. These are fundamental aspects of NMR spectral analysis.

We are grateful to the reviewer for the insightful feedback, which has greatly improved the quality of our revised manuscript. To ensure the reliability of our findings, we present NMR data from one representative experiment conducted on $n = 2$ biologically independent samples as the mean values \pm SD for the following experiments:

1. 50 μ M dN36-MepS in the absence and presence of 25-50 μ M NlpI dimer (**Supplementary Fig. 2**);
2. 50 μ M mMepS in the absence and presence of 25 μ M NlpI dimer (**Supplementary Fig. 3**);
3. 50 μ M MepS-L24R variant in the absence and presence of 25-37.5 μ M NlpI dimer (**Supplementary Fig. 12**);
4. 50 μ M MepS-F31A variant in the absence and presence of 25-100 μ M NlpI dimer (**Fig 4G**);
5. 50 μ M mMepS in the absence and presence of 50-100 μ M Prc^{SK} (**Supplementary Fig. 14**);
6. Heteronuclear [¹H]-¹⁵N NOEs of mMepS measured at 800 MHz (**Supplementary Fig. 5**).

2. For other experiments, such as SPR and ITC, basic control experiments were not conducted in the previous versions. Critical information, such as whether the protein was lipidated and which specific domain or mutant was used, is still missing. Instead, the text provides some irrelevant details.

We agree with the reviewer's assessment and apologize for the misleading results for SPR experiments in the initial version of our manuscript. In the revised manuscript, we added information about the protein constructs and made efforts to eliminate irrelevant details. The soluble mature forms of MepS (mMepS) and NlpI proteins were expressed and purified without the lipoprotein signal peptides. The DNA sequence encoding mature MepS (residues 2-162, corresponding to residues 28-188 in the MepS precursor) was cloned and inserted into either pET21a or pET28a vectors, which express a C-terminal His-tag or an N-terminal His-tag and a TEV cleavage site, respectively. The lipidation of MepS, which is located at residue C1 in the mature form of MepS, has been replaced with a Met residue. NlpI (residues 20-294) was cloned into pET28a vector with an N-terminal His-tag and a TEV cleavage site, and purified mature NlpI was soluble up to 200 mg/ml in 10 mM Tris/HCl at pH 8.0 and 10 mM NaCl. The lipidation of NlpI, occurring at residue C19, was not included in our construct.

3. Line 169: Supplementary Figure 4. The spectra superimposition cannot conclusively indicate the same oligomeric state. To determine the oligomeric state from NMR data, the average T_1/T_2 relaxation time ratio or, at the very least, the peak linewidth should be measured. The statement "the peak linewidth was pretty similar" lacks precision. If the linewidths have been measured, the values should be indicated in the figure.

Thank you again for your comments. To clarify the issue, we performed 1D ^{15}N -edited relaxation experiments to measure the average ^{15}N T_1 and T_2 relaxation times for mMepS and dN36-MepS. The data were acquired on a Bruker 800 MHz spectrometer at 298 K using pseudo-2D ^{15}N T_1 and T_2 gradient experiments. T_1 spectra were acquired with delays, $T = 20, 50, 100, 200, 300, 400, 600, 800, 1000, 1200,$ and 1500 ms, and a relaxation delay of 3 s. T_2 spectra were acquired with CPMG delays, $T = 16, 32, 48, 64, 80, 96, 128, 160, 192, 240,$ and 320 ms, and with a relaxation delay of 1.5 s. To minimize contributions from the disordered regions of mMepS, ^{15}N T_1 and T_2 values of mMepS and dN36-MepS were extracted by the decay of the integrated $^1\text{H}^{\text{N}}$ intensity between 9.0 and 9.6 ppm and the data were fitted the curves with standard exponential equations using the program 't1guide' within Topspin 4.0.6. We obtained a τ_c value of 8.7 ns for mMepS, indicative of a monomer, while dN36-MepS exhibited a τ_c value of 8.5 ns.

4. Line 275: A variation of $0.09 + 0.17$ is not significant. The error is very high. The authors should consider the variation of intensity above the standard deviation. This is valuable for all intensity variation measurements. The whole paragraph is not very clear.

We thank the reviewer for this suggestion. In the presence of 25-37.5 μM NlpI dimer, the NMR peaks of the L24R variant (50 μM) almost disappeared, except for a few residues close to L24R, resulting in a high error for the mean ratio of NMR peak intensities for the N-terminal region. To avoid misunderstanding, we have removed the statement regarding the mean ratio of NMR peak intensities for the N-terminal region. In the revised manuscript, we present error bars in the intensity ratios representing the standard deviation of $n = 2$ independent biological samples. We have also revised the main text and legends accordingly.

In NMR titration experiments, the binding site can be determined based on the location of the most significant shift or intensity changes. However, there is no universally defined criterion for defining significant changes, and users have the flexibility to set their own threshold. Typically, users calculate the standard deviation of the changes and set the threshold at 1 or 2 times the standard deviation. Based on the interaction between ^{15}N -labeled 50 μM mMepS and unlabeled Prc^{SK}, the peak intensity of mMepS did not significantly change, reaching $93\% \pm 11\%$ in the presence of the same concentration of Prc^{SK}. Even after titration with twice the amount of unlabeled Prc^{SK}, the NMR signal of mMepS remained at $91\% \pm 11\%$, suggesting that no significant binding was detected between mMepS and Prc^{SK}. Therefore, an I/I_0 ratio of 0.8 for mMepS in the NMR titration experiments is considered significant. The black dashed line represents 80% signal intensity across all the titration results.

5. I am not convinced by the explanation on the stoichiometry in the ITC experiments. The authors explain that the NlpI dimerization interface is crucial for mMepS binding. However, they use a mutant of NlpI unable to dimerize. The use of this mutant is not mentioned in figures and their legends, nor in the method section, even in the revised manuscript.

We appreciate your comments. We have endeavored to elucidate the stoichiometry in the ITC experiments to the best of our ability, albeit acknowledging that it may not be flawless. In the revised manuscript, we have included information about the monomeric mutant NlpI- Δ N (Supplementary Fig. 13A). NlpI- Δ N (37-294) mutant, which lacks residues 1-36, is unable to form a dimer and adopts a monomeric conformation. The SEC data are presented for dimeric NlpI (residues 20-294) and monomeric NlpI- Δ N (residues 37-294), represented in grey and red, respectively (Supplementary Fig. 13A). In accordance with the NlpI-mMepS complex structure, the N-terminal region of one protomer (mMepS-1) prominently engages with the dimerization interface of NlpI while that of the other protomer (mMepS-2) only interacts with one of NlpI. This suggests that the dimerization interface of NlpI is crucial for the binding of mMepS-1 but not mMepS-2 (Supplementary Fig. 8). Titration of unlabeled NlpI- Δ N into ^{15}N -labeled mMepS caused significant changes in the peak intensity from residues 26-162 (Supplementary Fig. 13B). The binding of NlpI- Δ N to mMepS was directly observed via NMR experiments. However, the interaction between mMepS and NlpI- Δ N did not exhibit exothermic or endothermic signals, as monitored via ITC (Supplementary Fig. 13C). Therefore, we propose that only mMepS-1 interacts with NlpI dimer with an enthalpically favorable binding response as mMepS-2 could bind to NlpI without apparent heat release or absorption; therefore, a lower stoichiometry was observed in the ITC binding reaction.

6. The order of supplementary figures is not aligning with their references in the main text, causing difficulty in reading.

Thank you for pointing out the issue. We have revised the manuscript accordingly.

7. In general, the figures and the legends are not clear, the color codes and abbreviations are missing in the legend. A careful analysis of figures and their legends is required.

We express our gratitude for bringing the issue to our attention. Subsequently, we have made revisions to the manuscript accordingly.

Reviewers' Comments:

Reviewer #3:

Remarks to the Author:

The authors have improved their manuscript by additional explanations that were really missing to well understand their conclusions.

The figures of NMR spectral superimposition are now well presented and analysed.

While I remain unconvinced by the explanation regarding the ITC stoichiometry, I appreciate that an explanation has been provided this time. Readers and experts will have the opportunity to form their own judgments.

The manuscript is now acceptable for publication.